# Learning Formal Mathematics
# From Intrinsic Motivation

**Gabriel Poesia**[1]    **David Broman**[4]    **Nick Haber**[1,3]    **Noah D. Goodman**[1,2]

{poesia,nhaber,ngoodman}@stanford.edu    dbro@kth.se

Departments of Computer Science[1], Psychology[2] and Education[3], Stanford University, USA
EECS and Digital Futures[4], KTH Royal Institute of Technology, Sweden

## Abstract

How did humanity coax mathematics from the æther? We explore the Platonic view that mathematics can be discovered from its axioms—a game of conjecture and proof. We describe MINIMO (Mathematics from Intrinsic Motivation): an agent that jointly learns to pose challenging problems for itself (*conjecturing*) and solve them (*theorem proving*). Given a mathematical domain axiomatized in dependent type theory, we first combine methods for constrained decoding and type-directed synthesis to sample valid conjectures from a language model. Our method guarantees well-formed conjectures by construction, even as we start with a randomly initialized model. We use the same model to represent a policy and value function for guiding proof search. Our agent targets generating hard but provable conjectures—a moving target, since its own theorem proving ability also improves as it trains. We propose novel methods for hindsight relabeling on proof search trees to significantly improve the agent's sample efficiency in both tasks. Experiments on 3 axiomatic domains (propositional logic, arithmetic and group theory) demonstrate that our agent can bootstrap from *only* the axioms, self-improving in generating true and challenging conjectures and in finding proofs.

## 1 Introduction

Mathematical reasoning stands as a grand challenge for Artificial Intelligence (AI) research since the birth of the field [25]. Artificial agents capable of general mathematical reasoning have the potential to drastically impact both mathematics itself and areas where mathematical proof plays a key role, such as program and hardware verification [4]. While this goal has received significant attention from the AI community [22, 46, 30, 42], it still remains far from the breakthroughs that areas such as general game playing [36], image [32] generation or protein folding [35] have witnessed.

Prior work has reflected two main visions of how AI might achieve general mathematical reasoning abilities. One such strategy is to leverage all of the available human-produced mathematical knowledge as a starting point [31]. This knowledge is encoded in source as varied as textbooks, online forums, academic papers, as well as formal proofs written in computer languages such as Lean, Isabelle, Coq or Metamath [2]. Large Language Models (LLMs) can ingest all such sources of knowledge in a unified manner, and provide a foundation for tasks in both formal and informal mathematics. Benchmarks of mathematical problem solving in natural language, such as GSM8k [10] and MATH [15], have measured rapid progress over the years, but they remain limited to problems where the final answer is a number, due to the challenge of assessing the correctness of mathematical *arguments* written in natural language. This difficulty is not a challenge in formal theorem proving, where we can automatically verify the correctness of *proofs* in arbitrarily high-level mathematics. But benchmarks of formal theorem proving (such as minif2f [46] and LeanDojo [44]), even with rapid advances in LLMs, have not yet witnessed the same breakthroughs. In fact, these benchmarks remain

38th Conference on Neural Information Processing Systems (NeurIPS 2024).

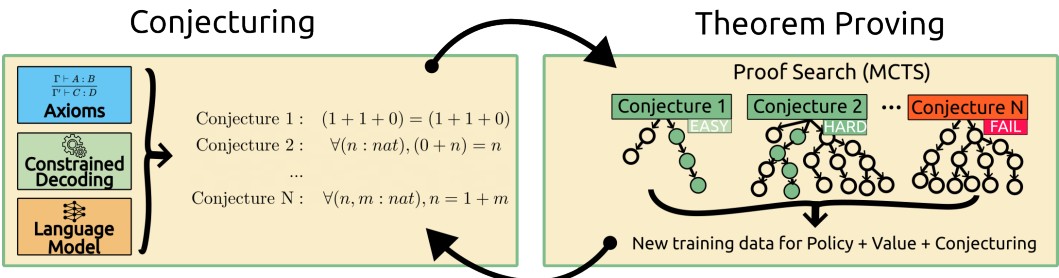

Figure 1: We train mathematical reasoning agents starting only from the axioms of a given formal domain, where they jointly learn to pose challenging but provable conjectures and to find their proofs.

far from solved even though all of the theorems and problems in them are known mathematical facts, often already presented informally in publicly available training data.

An alternative vision of how AI might master mathematical reasoning sees mathematics through the lens of game playing, observing that the rules of the "game of mathematics" can be encoded in a variety of formal systems, such as dependent type theory, using a small set of axioms [24]. In principle, this can allows us to potentially borrow the successes of general game-playing agents, such as AlphaZero [36], that have achieved remarkable success in mastering complex games entirely from experience. Notably, AlphaZero achieves super-human performance in a range of games without leveraging any human examples. Instead, it learns entirely from experience given only an environment where it can play the game by itself. If this approach could be transported to mathematics, it would bypass the dependency on human examples, and allow us to explore mathematical domains — both known and new, without distinction — by utilizing large scale compute and the potential of axioms to produce infinite data.

However, there is a crucial and often neglected difference between mathematics and traditional board games where game-playing AI has succeeded: mathematics is a game with *intrinsic rewards* [8]. Board games, such as Go or Chess, have a fixed starting configuration, and their rules determine the outcome of the game unequivocally. Mastering the game then amounts to learning a policy that optimizes for the *extrinsic* signal of winning. In theorem proving, a starting configuration is given by a theorem statement, and the correctness of a proof can be assessed objectively in a formal system. But the choice to work on a particular statement — a *conjecture*, before it is proved — is not given a priori by the rules of the game of mathematics. Instead, these goals come from the *intrinsically motivated* agents [8, 34] who are playing the game. Thus, a key skill developed by human mathematicians is to decide which conjectures are worth considering. In stark contrast, current benchmarks of mathematical reasoning abilities, both formal [46, 44] and informal [10, 15], all measure performance on an extrinsically defined set of problems, without space for further discovery.

In this paper, we make the first step towards creating *intrinsically motivated* mathematical agents by proposing MINIMO — Mathematics from Intrinsic Motivation —, a method to jointly learn conjecturing and theorem proving, starting from *nothing but the axioms of a given mathematical domain*, represented in dependent type theory [12]. We borrow inspiration from the literature of *intrinsic motivation* in Reinforcement Learning (RL) [8, 7, 33, 26], where agents can learn from interacting with an environment even when no specific goals are given. Intrinsic motivation objectives have been instrumental for RL in hard exploration environments, where rewards are too sparse to seek directly [5]. The sparsity of rewards is also a major challenge in theorem proving, making this connection especially attractive. We thus define the objective of conjecturing as generating new problems that are challenging for the current agent but still provable within its given search budget. Since the agent also learns from the solutions it finds, the conjecturer has to continuously increase the difficulty of the problems it generates.

MINIMO performs both conjecturing and proof search with a Transformer [40] language model (LM), which starts randomly initialized. To sample conjectures even from a model that starts with no prior knowledge, we combine methods from type-directed program synthesis and constrained generation from language models, enabling us to get valid conjectures *by construction* — concretely, conjectures are simply terms of type `prop`, the type of mathematical propositions in our type theory. Then, we perform proof search in the Peano environment [28], which provides a finite action space for search

in a dependent type theory, guiding search using the LM as a policy and value function. When a proof is found, proof search generates training data for improving the policy and values; it also provides data to improve conjecturing, since we then know how hard the problem was. We use this to alternate between conjecturing and theorem proving, in a self-improving loop. However, successful proofs are sparse. We thus adapt the idea of *hindsight relabeling* [1] to reinterpret failed trajectories as successful ones by rewriting their goals a posteriori. This significantly accelerates learning, allowing us to extract hundreds of new (true) conjectures, and their proofs, even from failed proof searches. In this way, even unprovable conjectures can be highly useful for the agent's learning. We evaluate our system on three axiomatic mathematical domains—propositional logic, natural number arithmetic, and group theory—showing both that agents self-improve successfully in proving theorems in all domains. We also find that they improve at an extrinsic evaluation of theorems (from a classical textbook on logic [20], and the Lean Natural Number Game [6]), *not given in training*. In summary, we make the following contributions:

- We introduce a method for conjecturing using LMs that generates valid conjectures by construction in an arbitrary theory, using constrained decoding and type-directed synthesis.
- We define a hindsight relabeling method that simultaneously generates training data for *conjecturing* and *theorem proving*.
- We combine these methods in a learning loop where a mathematical agent can self-improve in a given formal domain given *only* the axioms.
- We evaluate agents trained on axioms for propositional logic, group theory and arithmetic, showing that they improve in both intrinsic and extrinsic evaluations.

## 2   Related Work

Our work is primarily related to prior work on mathematical conjecturing, learning to prove theorems, and on intrinsic motivation in Reinforcement Learning. To the best of our knowledge, our work is the first attempt to unify insights from these areas for training mathematical reasoning agents.

**Mathematical conjecturing.**     The task of *discovering* mathematical facts was the subject of the influential Automated Mathematician (AM), developed by Lenat in the 1970s [23]. AM was able to conjecture several known mathematical facts and concepts (such as the definition of prime numbers, and the unique factorization theorem). Unlike our system, AM did not aim at *proving* the conjectures it formulated — instead, it proposed and judged them based on a set of principles and on empirical evidence collected by AM itself. More recently, other works have revisited the idea of generating conjectures by training language models on human-written theorem statements [39, 41]. Unlike our approach, this relies on pre-training data, and does not readily extend to conjecturing in new domains.

**Learning to prove theorems from human data.**     A large body of recent work has used Large Language Models to guide formal theorem proving in a number of proof assistants, such as Lean [22], Isabelle [22, 18] and Metamath [31, 42]. Typically, these systems pre-train an LLM on large-scale Internet corpora and fine-tune on human-written formal mathematical proofs. Work on scaling laws for LLMs has shown that they generally improve when their parameter count and dataset size both increase in similar proportion. But the scarcity of formal proofs for training creates a challenge for this approach for learning formal theorem proving: even the largest datasets to date, such as the ProofPile, which aggregates libraries from 5 proof assistants, form relatively small datasets (e.g., 500MB of formal proofs on the ProofPile, contrasting to terabytes of Python code on Github).

**Learning to prove theorems from synthetic data.**     One recent success in automated mathematical reasoning was AlphaGeometry [38], which was highly effective in solving olympiad-level geometry problems. AlphaGeometry, like our method, was trained entirely on synthetic data. Crucial to its approach is a method for generating both problems and solutions using only the axioms of geometry and a domain-specific deductive closure solver. This allows AlphaGeometry to synthesize and train on hundreds of millions of problems: many orders of magnitude more than existing human-created datasets of mathematical problems. Our approach shares the goal of AlphaGeometry of only using data derived from the axioms of the domain, with the difference that our method (a) is agnostic to the underlying axiomatic system and (b) does not rely on a domain-specific solver. Another line

of work, including TacticZero [43] and rlCoP [19], has explored learning to prove theorems from reinforcement learning only, in a tabula rasa fashion, but still using a human-written set of problems for training (and manually-engineered features, in the case of rlCoP).

**Intrinsic motivation** We leverage inspiration from the literature on training reinforcement learning agents with intrinsic motivation objectives, allowing an agent to learn without pre-specified goals [26, 34, 33, 13, 27, 7, 3, 37]. Our setup is conceptually close to AMIGO [7], where agents attempt to generate challenging but achievable next goals. While AMIGO was demonstrated in a simple grid-world environment with a simple goal structure (any point in the grid gives a valid goal), we operate on a much richer setting, where the space of goals is unbounded — all conjectures in a formal mathematical theory. To sample conjectures, we use Synchromesh's Constrained Semantic Decoding algorithm [29], and guide it with type constraints.

**Self-improvement of language models** A significant line of recent work has explored the idea that LLMs can "self-improve": increase performance on a given task by fine-tuning on their own generated reasoning traces, which are selected by some criterion that ensures their quality. STaR [45] fine-tuned on reasoning traces that reached the correct answer on mathematical and multiple choice questions; LMSI [16] was able to obtain self-improvement on a question-only dataset, sampling multiple rationales and training on those that agree with the majority answer. More related to our work, but less explored, is the direction of having LLMs also generate their own *problems* for training: this has been explored for self-improving on solving programming problems [14, 37], where code execution provides signal about correctness. In MINIMO, since our problems are formal conjectures and our solutions are formal proofs, we can guarantee correctness in a much stronger form than self-generated test cases.

## 3 MINIMO

Most recent work on AI for mathematical reasoning assumes a target set of problems to be solved. We deviate from this paradigm by having *the agent itself* propose problems for it to try to solve and learn from. Our goal is to target increasingly harder problems in a given mathematical domain, where the domain is specified as a set of axioms given in dependent type theory.

Our agent is represented by a language model, which we will use to encode (a) a proof search policy $\pi_\theta(a|s)$, (b) a value function $V_\pi(s)$, and (c) a difficulty-conditioned *conjecturer* $P_\theta(c \mid d)$, where $d$ is a discretized measure of difficulty and $c$ is a mathematical statement (a string). MINIMO consists of training both components in a loop that alternates between generating conjectures, trying to target hard but provable ones, and doing proof search, as we depict in Figure 1. As we describe in this section, proof search generates training data both for the conjecturer and the prover components — training on that data thus yields a self-improvement loop as the agent interacts with the environment. We now describe the first step in this loop: generating conjectures.

### 3.1 Conjecturing

We aim to sample conjectures from a language model, conditioned on a target difficulty. By construction, an autoregressive LM gives a distribution over all strings. But if the LM does not have prior knowledge about the domain, it is unlikely to put non-negligible probability mass on valid mathematical statements. We now address the challenge of sampling valid conjectures.

To that end, our main insight is to leverage *constrained decoding* to obtain valid conjectures by construction. Our method will turn any language model — including a randomly initialized one, which is what we start with — into a probability distribution over *strings that represent well-formed conjectures* in dependent type theory over a given set of axioms. Ultimately, we will also train the LM to generate conjectures given a target difficulty. We use this ability to attempt to generate increasingly difficult problems for training, according to the agent's current ability to prove theorems.

To reason about constraining the LM's outputs, we leverage the abstraction of a *completion engine*, first introduced in the context of code generation with language models [29]. Assuming $\mathcal{C}$ is the set of all valid conjectures, a completion engine will allow us to use any LM to sample strings from $\mathcal{C}$ in an autoregressive fashion. Mathematically, $\mathcal{C}$ is a function $f_\mathcal{C} : \Sigma^* \to \mathcal{P}(\Sigma^*)$, taking a string $s \in \Sigma^*$ and

returning a set of strings $f_\mathcal{C}(s)$[1]. Intuitively, we will sample conjectures from our LM by constraining it to strings that can be generated with iterated calls to $f_\mathcal{C}$. Concretely, we will start with $s =$ "", and query $f_\mathcal{C}(s)$ to obtain the set of valid ways to begin to state a conjecture. After we choose one of those $s_1 \in f_\mathcal{C}(s)$, we can then query $f_\mathcal{C}(s_1)$ to obtain the valid ways to proceed, and repeat until we have a complete conjecture. Our main challenge here is to construct a suitable $f_\mathcal{C}$ that it is *sound* (all conjectures obtained by this procedure are valid) and *complete* (all valid conjectures can be obtained by this procedure). After we define $f_\mathcal{C}$, we can sample from any LM while guaranteeing that the output will belong to $\mathcal{C}$ by using the Constrained Semantic Decoding (CSD) algorithm [29].

To construct $f_\mathcal{C}$, we analyze the minimal formulation of dependent type theory backing Peano [28] – essentially the classical Calculus of Constructions (CoC) [12]. In the CoC, mathematical propositions are constructions of type prop. Since generating a proposition might involve generating objects of arbitrary other types, depending on the axioms of the given domain, we will define a more general family of completion engines, $f_t$, which will constrain the LM to sample an object of an arbitrary type $t$. At the end, we obtain $f_\mathcal{C} = f_{prop}$.

To sample an object of type $t$, we leverage the simple grammar of terms in Peano [28] to guide a recursive type-directed synthesis algorithm. Syntactically, terms in Peano are defined by the grammar $e ::= \text{type} \mid \text{prop} \mid x \mid (e\ e) \mid (\text{lambda } x : e, e) \mid [(x : e) \to e]$. The first two production rules give the names of two built-in base types, type and prop. We then have variables, function application, lambda functions, and function types (denoted in square brackets). As conventional in type theory, let $\Gamma$ be our *typing context*: a sequence of pairs of names and their types. For example, we might have $\Gamma = [nat : type, z : nat, succ : [nat \to nat]]$, a context with three objects: a type nat, an object z having that type, and a function succ from nat to nat. Given a context, to obtain an object of type $t$, it suffices to consider the formation rules in CoC to guide generation:

- If our target type is $t = \text{type}$, we can generate either one of the names in $\Gamma$ having type $t = \text{type}$ (e.g., nat, for the example context above), or a function type.

- If our target type is $t = \text{prop}$, we can generate either one of the objects in $\Gamma$ having type prop, or a function type where the output type has type prop.

- If our target type is a function type, we must start by generating a square bracket; then, we (recursively) iteratively generate a type for the next parameter, or, if we already have at least one parameter, we can also generate the final output type.

- An object of any type $t$ can be formed by function application of a function $f$ chosen from $\Gamma$, provided that the output type of $f$ can be unified with $t$.

These rules allow us to determine the possible valid tokens at any point during generation. Besides the base types type and prop, objects of all other types are either in $\Gamma$ or the result of a function application. We use a recursive descent parser to parse the (incomplete) term we have so far (as originally done in [29]), and compute the target type $t$ at the current point in generation. Then, we enumerate the possible next tokens for the LM by using the rules above, return the union of the sets of choices allowed by each rules as the output of $f_t$.

Note that this is a general procedure for searching for objects of any given type $t$ (i.e., *inhabitants* of that type). This is undecidable in general (theorem proving is a special case of type inhabitation), so this procedure might not terminate. Therefore, we set a maximum number of tokens $K$ (150, in our experiments), and ignore samples where the LM fails to generate a conjecture after $K$ tokens. In practice, we find the rejection rate for generating *propositions* to be low, $< 10\%$ of the generations.

The above procedure forces the LM to generate a *valid* conjecture, but the LM still assigns a distribution over those. During training, we also aim to generate conjectures that are provable, but hard to prove. The signal on both success and difficulty is generated by running proof search (as we describe next) and, in cases where a proof is found, measuring it's log-likelihood under the current policy, which correlates with how many iterations MCTS takes to find the proof (see Appendix).

---

[1]This set can be implicitly defined by a regular expression, so it might be infinite. This allows a completion engine to represent constraints such as "what follows can be any valid identifier".

## 3.2 Proof Search

Having a conjecture represented by a target type $t$, we then perform proof search using Monte Carlo Tree Search (MCTS; [21], [42]), guided by a learned policy $\pi_\theta$ and value function $V_\theta$. We represent both $\pi_\theta$ and $V_\theta$ using the same underlying language model that we use for conjecturing. We use Peano [28] as the environment for proof search. Peano exposes a finite action space, so we don't need to *generate* actions using $\pi_\theta$ — it suffices to be able to evaluate them. More precisely, at a given state $s$ where we have actions $a_i^{(s)}$ available, we compute the distribution $\pi_\theta(a_i^{(s)}|s)$ by evaluating the likelihood of each $a_i^{(s)}$ as the completion to a string of the form `"STATE: «s»; POLICY:"`. We read out the value of a state in a similar way — by considering the likelihood of $1$ or $0$ as being the next token following `"STATE: «s»; VALUE:"`. In both cases, the probability of the next token is normalized over only the choices that can lead to a valid completion.

States and actions in Peano are similar to several other interactive theorem provers, such as Lean and Coq. The state consists of a set of typed objects, along with a set of open proof goals (which are types). Objects in the state whose type is a proposition type are interpreted as evidence for that proposition (either a proof or an assumption). Actions might add new objects to the state (e.g., take the successor of one existing natural number, or use an equality to rewrite an existing proposition into a new one), change the goal (by backward chaining), or both (e.g., if the goal is to prove $A \rightarrow B$, which is used to represent both logical implication and universal quantification, the `intro` action will add object of type $A$ to the state and change the goal to $B$).

When proof search succeeds, we can extract examples to train both the policy and the value function. For the policy, we simply extract the actions that lead to the proof as language modeling examples, using the same format we use to query $\pi_\theta$. For the value function, we take the states in the subtree that complete the proof as examples with value $1$, and a random set of other states as examples with value $0$ for training. When proof search fails, however, this naïve procedure does not extract any training examples. This happens often at the beginning, since our model initially generates a large number of conjectures it cannot prove (either because they are false, or because naïve proof search times out). But forward actions in the proof search tree often construct proofs for other propositions, even if they are irrelevant for proving the original goal. We levarage this fact for generating training data by hindsight relabeling, as we describe next.

## 3.3 Hindsight Relabeling

Even a conjecture that fails to be proven can be highly informative about the domain. During proof search, forward actions that apply functions whose result type is a proposition type (e.g., concluding `A` from `(and A B)`) produce proofs, even when those proofs might not be useful for proving the original conjecture. In Reinforcement Learning, the well-known method of Hindsight Experience Replay [1] extracts training data for the policy from such failed trajectories by relabeling the trajectories with goals that were in fact achieved, as opposed to the original goal. For those alternative goals, the trajectory then represents a successful sequence of actions. We apply this idea to extract training examples for both the policy and value functions from proof search trees, by picking nodes after forward actions that produced a proof, and walking upwards in the tree until we find a backward action (since those change the goal). That path then becomes a successful trajectory after we relabel the goal. Two important steps to improve data quality are (1) we clean up the solutions by eliminating steps irrelevant for the proof of the new goal, and (2) we only add proofs of goals never seen before, to avoid oversampling trivial facts that are rediscovered extremely often (such as $0 = 0$).

We go one step further and observe that hindsight relabeling can also be useful for training the *conjecturer*. Concretely, the procedure we described above produces a set of proofs $p_i$ for relabeled statements $g_i$. All of these statements are therefore true in the mathematical domain, and we use them as examples of true conjectures. As a concrete example, in arithmetic, the agent will often conjecture simple expressions such as $2 + 1 = 0$. Most equalities generated at random will be false. However, by applying the Peano Axioms and facts about equality in succession, the agent eventually finds a proof of $2 + 1 = 3$. Evaluating the likelihood of the proof under $\pi_\theta$ gives a measure of the difficulty of this alternative statement for the current policy. This insight allows our agent to learn about hundreds of new unique, true statements in each proof search. As our experiments show, we find hindsight relabeling to be crucial for enabling the agent to steadily conjecture harder statements that it is able to prove.

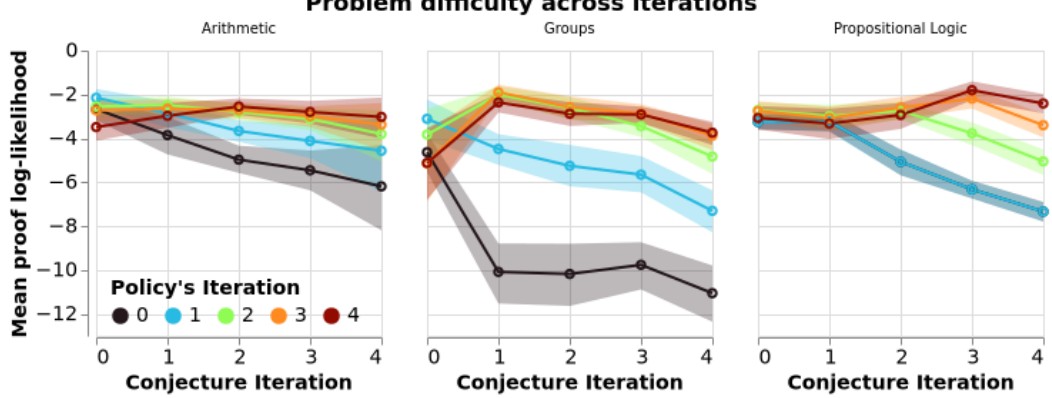

Figure 2: Difficulty of proved conjectures found in each iteration of MINIMO, evaluate as the log-probability of the proof under the policy checkpoints after each iteration of the training loop (with standard error bands for runs with 3 random seeds).

## 3.4  Training loop

Finally, we tie all components together by alternating between (a) generating a batch of conjectures, (b) running proof search on them, (c) extracting training examples from the search trees, and finally (d) training the underlying LM using the standard cross-entropy objective. When a proof is found, either directly or by hindsight relabeling, we first compute a continuous difficulty metric of the statement by taking the log-likelihood of the proof under the current policy. To discretize difficulties, we then consider difficulty percentiles of the last batch of conjectures: we take the 10% least likely proofs to be associated with "hard" conjectures, the 50% most likely to be considered "trivial", and the remaining are taken as "easy". When put together, these components form a self-improving loop that starts only from the axioms of the given mathematical domain, as our experiments show.

## 4  Experiments

We now evaluate MINIMO[2] on three mathematical domains. We experiment with axiomatic systems for (a) propositional logic, (b) arithmetic, and (c) abstract groups. All the axioms are given in the Appendix. We then train agents over 5 iterations of conjecturing and theorem proving, generating 200 conjectures in each batch, running MCTS for proof search with 1000 expansions, and evaluate our agents guided by the following research questions:

**RQ1:**  Can our conjecturing method generate increasingly harder conjectures as training progresses?

**RQ2:**  Do agents improve at theorem proving as they train on their own generated problems?

**RQ3:**  Is hindsight relabeling effective at improving conjecturing and theorem proving?

**RQ4:**  Do our agents improve at proving an externally given set of human-selected theorems, even if these are not given during training?

The first three questions require *intrinsic evaluations* — they ask whether the learning dynamics of agents trained with MINIMO matches the desiderata of self-improving at both conjecturing and theorem proving while given only the axioms. The last question is an extrinsic assessment of what our agents learn — we evaluate whether the learning progresses in a way that aligns with an external criteria — namely, the proficiency of the agent at proving theorems from human sources (a textbook and a Lean game).

### 4.1  Learning dynamics

We first investigate RQ1 and RQ2. Figure 2 shows the progression of difficulty across 5 iterations of the MINIMO learning loop (first conjecturing, then running proof search, and training on collected

---

[2]Our code is available at `https://github.com/gpoesia/minimo`.

examples). Here, we evaluate the average log-likelihood (y-axis) of conjectures proven at each conjecturing iteration (x-axis) under the policy after each iteration (line color). Policy 0 is the initial (randomly initialized) policy, while subsequent policies were trained on the examples obtained during proof search, with hindsight relabeling, up to the previous iteration. Lower log-likelihood generally means harder conjectures (i.e., they tend to take more search iterations, see Appendix).

**Difficulty increases as training progresses (RQ1).** Fixing an early policy, the log-likelihood of proofs under that policy steadily decreases across training iterations. This is reflected in the negative slope of difficulty across iterations when the policy is fixed. In particular, the policy at iteration 0 serves as a naïve search baseline, since it is essentially uniform. We observe that, as training progresses, this policy struggles increasingly more with each new batch of conjectures. The same pattern repeats for subsequent policies when we consider conjectures generated in future iterations, giving a generally downward trend in log-likelihood of the solutions for any given policy, showing that conjectures get increasingly more challenging. This provides positive evidence for answering RQ1: in all 3 domains, the conjecturer is able to increasingly propose harder provable conjectures as training progresses.

**Proof policy improves as training progresses (RQ2).** At each iteration, we sample a new set of unseen conjectures for training. From Figure 2, we see that later policies generally perform better than earlier ones, at a fixed conjecture iteration. This reflects the fact that each new policy assigns higher likelihood to the proofs, even for unseen conjectures at each iteration. For example, after training on the data generated from the first iteration, the policy on iteration 1 has higher log-likelihood for the proofs to the *new conjectures* found at iteration 1, when compared to the initial policy from iteration 0. This pattern repeats at each iteration, showing that the prover is also progressing and generalizing to unseen problems, though with diminishing gains in the final iterations. This suggests a positive answer to our second research question: our agents effectively self-improve in their ability to prove new statements.

**Hindsight relabeling is necessary for joint self-improvement (RQ3).** The results so far all used hindsight relabeling on all proof searches—successful or not—to extract training data for the policy and value function, as well as conjecturing. To evaluate whether our agents would still show the same continuous self-improvement regardless of the data efficiency gains from hindsight relabeling, Figure 3 compares agents trained with and without hindsight relabeling across 5 iterations over the same 3 axiomatic domains. Here, we look at the ability of the agent to achieve the goal of constantly proposing provable but challenging conjectures for itself. Ideally, the difficulty of conjectures should not fluctuate significantly during the course of training: we would like the agent to always find new challenging conjectures that it nevertheless still proves. We find that, in all 3 domains, the agent fails to achieve this goal when not trained with hindsight relabeling. Instead, as it trains on its own proofs, the agent's conjectures fail to remain challenging—all provable conjectures end up with high log-likelihood as training progresses, and the conjecturer is unable to leave that regime. We attribute this to the volume of signal that the conjecturer receives: at each initial batch, only around 10-20% of the conjectures are proven. When not using hindsight relabeling, the only feedback that the conjecturer recevies is that proof search timed out on these statements. On the other hand, with hindsight relabeling, even these failures lead the conjecturer to observe hundreds of actual true statements in each domain (along with their proofs), leading to better learning dynamics. This provides positive evidence for RQ3: hindsight relabeling significantly helps the agent to jointly improve in conjecturing and theorem proving—without it, training tends to collapse to by proposing only easy conjectures.

## 4.2 Proving human-written theorems (RQ4)

Finally, we evaluate whether our agent, trained only on problems that it proposes to itself, also improves in solving problems that are of interest to humans. Since our agent does not grow its library of theorems over time, starting every new proof from the axioms, a meaningful evaluation requires theorems that can be reasonably proven straight from the axioms, without lemmas. We thus use two human-written sources of theorems meeting this criterion, for the domains of propositional logic and arithmetic. For logic, we take the set of 35 theorems of propositional logic from Stephen Kleene's textbook "Introduction to Metamathematics" [20]. Precisely, in Theorem 41, Kleene states (and proves a subset of) 35 useful statements of Propositional Logic (such as contraposition

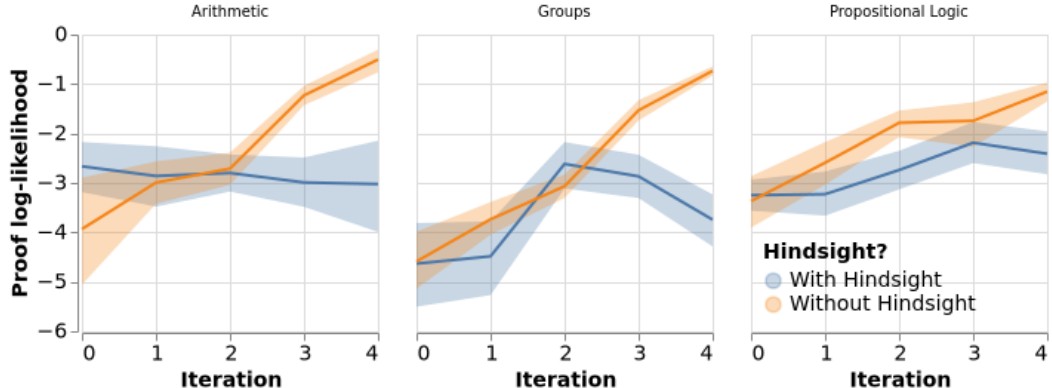

Figure 3: Difficulty of proved conjectures proposed in each iteration under the current policy at that same iteration, comparing when using and not using hindsight relabeling to generate new proofs and conjectures, with standard error bands for runs with 3 random seeds. Ideal behavior would be a flat line, representing constant relative difficulty. Hindsight significantly helps the agent conjecture propose harder problems.

rules, commutativity and transitivity laws of logical connectives, and properties of double negation). For arithmetic, we use the Natural Number Game [6], a popular game used to introduce formal mathematics in the Lean theorem prover. We take levels of the game that are (a) theorems about natural numbers, and (b) do not refer to previous lemmas, only the axioms; this results in 10 levels spanning the Tutorial, Addition, and Multiplication worlds. We translate the statements into Peano, and evaluate our agents on their success rate on those problems within 2000 MCTS expansions.

Figure 4 shows the results. We find that, as our agents train on their self-generated problems, they steadily become more successful at proving theorems from both Kleene's book and the Natural Number Game. This happens *even though these theorems are not targeted* during training, since our agent only uses its own conjectures. Four theorems in Propositional Logic are only proved after the last iteration of training: commutativity and transitivity of the "if and only if" logical connector, a law connecting double negation and implication ($\neg\neg(A \Rightarrow B), \neg\neg A \vdash \neg\neg B$), and the "currying law" of the conjunction — $(A \wedge B) \Rightarrow C \vdash A \Rightarrow (B \Rightarrow C)$. In the Natural Number game, only the final agent proves $\forall n, \forall m, succ(n) + m = succ(n+m)$, a theorem requiring induction on the correct variable (m) and a non-trivial sequence of rewrites in the inductive case (we include the full proof in the Appendix). While admittedly small scale, these results suggest a positive answer to our last research question: agents trained on their own conjectures can also improve at solving human-written problems, which are not given during training.

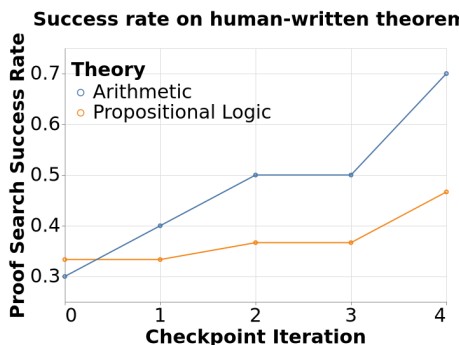

Figure 4: Success rate of our agents at proving theorems from the "Introduction to Metamathematics" textbook and the Natural Number Game. As agents train on their own conjectures, they also improve at solving problems from these two human-written sources.

## 5 Limitations and Conclusion

We present MINIMO: an approach to training agents for formal mathematical reasoning starting from only the axioms of a given domain. The agent jointly learns to propose challenging conjectures and to prove them. Our experiments show evidence of MINIMO improving its performance across training iterations. In the Groups domain, the average proof length it finds on *generated* conjectures (i.e.,

not found by hindsight) increased from 2.67 steps in the first iteration, when the model is randomly initialized, to 5 steps by iteration 4, with proofs for 'hard' conjectures growing from 3.67 to 6.10 steps. The longest proofs found grew from 4 steps to 9 steps from the first to last iteration. Similar trends appear in Propositional Logic (average length from 2.75 to 4.21 steps, longest proofs from 5 to 11 steps) and Arithmetic (average from 2.36 to 3.35 steps, longest from 4 to 7 steps).

However, MINIMO currently has two crucial limitations that prevent it from (a) discovering deep mathematical theories, and (b) scaling up to large theories. First, even if the agent's policy improves, its library remains fixed to the definitions and axioms that it starts with. Proofs that do not use lemmas (*cut-free* proofs in logic) can be exponentially longer than equivalent ones that do, and thus quickly grow beyond the reach of search. With this constraint, our agent most often finds harder conjectures by making the statements longer and more complicated. For example, in Groups, early conjectures include trivial statements like $e = e$; by the last iteration, the conjecture requiring the longest proof reads as $\forall g \in G$, if $e = (g^{-1})^2$ then $e^2 = e(e(e((g^{-1})^2)))$ (proved in 9 steps). In contrast, human mathematicians develop deep theories by accumulating results and definitions along the way, in such a way that even very high-level results can be described succinctly at the right level of abstraction. Understanding how to bootstrap such cumulative learning in an agent (e.g., exploring notions of *usefulness* or *interestingness*, several of which have been posited [4, 11]) will be a key direction for future work.

Another bottleneck in our current setup is that a large library can cause the current action enumeration algorithm in Peano to become prohibitively slow (a finite action space can still become intractable). A method that scales unboundedly should incorporate some form of *premise selection* [17]. In past work, premise selection has either been based on symbolic heuristics or in learning useful premises from human-written proofs. We believe that developing a premise selection method that *bootstraps* together with the other learned components will be as important as understanding how to grow the agent's library.

Together, lifting these limitations from our method might lead to a completely compute-bound, self-improving agent for formal mathematics capable of discovering deep mathematical theories starting only from basic axioms — the rules of the game of mathematics.

### Acknowledgments

We thank Daniel Selsam and Laetitia Teodorescu for insightful discussions on preliminary ideas related to this work, as well as the anonymous reviewers for their helpful feedback. This work was supported by the NSF Expeditions Grant with Award Number (FAIN) 1918771, and by NSF grant #2302701. GP was also supported by the Stanford Interdisciplinary Graduate Fellowship (SIGF). This work was also partially supported by the Wallenberg AI, Autonomous Systems and Software Program (WASP) funded by the Knut and Alice Wallenberg Foundation.

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

# A   Training details

We represent our agents with GPT-2-style character-level Transformer models totalling approximately 8.45M parameters each (8 layers, 8 attention heads, hidden size 512, feed-forward size 2048, vocabulary size 128, absolute positional embeddings, with maximum context size of 1024). After each training iteration, we train the LM for a fixed number of 2000 steps of the AdamW optimizer (learning rate of $1e - 4$) with a dynamic batch size of most 10000 characters (random examples are added until their padded sequence lengths add up to more than 10000). We found these parameters to generally lead to stable training across all runs, without divergence, and 2000 steps was enough to bring each iteration to convergence in training loss.

Our training runs (5 iterations of generating and proving 200 conjectures in each) were done on 2 machines with 5 NVIDIA A40 40GB GPUs each. Each run took from 8 to 16h on a single GPU, totalling 288 GPU hours for the runs underlying our main results.

# B   Proof log-likelihood and MCTS Expansions

Throughout the paper, we used the likelihood of the proof under the policy as a measure of difficulty. Figure 5 shows that this quantity is linked to the number of iterations that MCTS takes to find the proof. This allows us to estimate difficulty easily without running search for theorems that we have proofs for (such as those we find by hindsight relabeling).

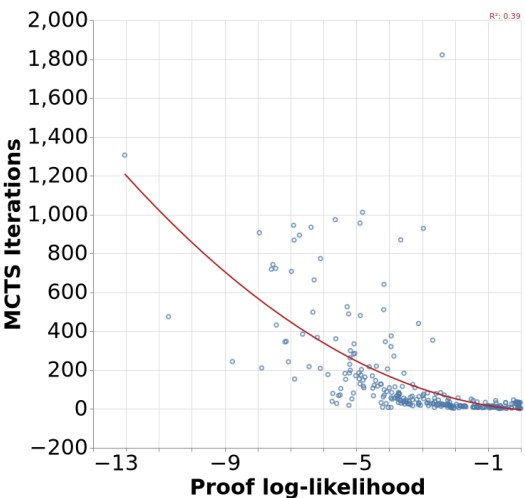

Figure 5: Relation between MCTS iterations until the proof is found, vs log-likelihood of the proof that is found. The higher the likelihood, the faster MCTS is in finding the proof.

# C   Fraction of provable conjectures across iterations

Our conjecturing procedure steers to LM to generate "hard", where that means they can still be proven by the current agent. Figure 6 shows how the *ratio* of proven conjectures evolves across training. In Groups and Propositional Logic, this ratio steadily increases both when and when not using hindsight. In Arithmetic, we observe that the ratio remains consistent up to iteration 3, but then *decreases*, whereas it *increases* sharply without hindsight. We note that this sharp increase is due to the conjecturer generating mostly trivial conjectures, by adding new combinations of assumptions that are not necessary for the proof (e.g., the most common conclusion for the conjecturer to generate without hindsight is often 0 = 0). Even though the hard conjectures with hindsight often fail to be proven, they still seem to generate useful training data for the prover, as we see in our extrinsic evaluation with problems from the Natural Number Game. Thus, this analysis points out that ideal behavior for the conjecturer is unclear — it is still perhaps positive to generate false conjectures, as long as the process of proving them leads to learning progress (as seems to be the case in Arithmetic).

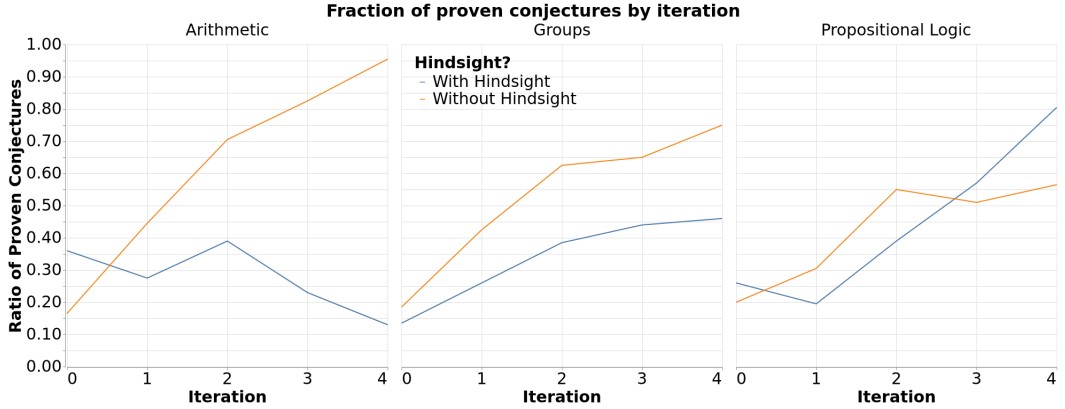

Figure 6: Ratio of proven conjectures in each batch of 200, at each iteration of training.

# D Axioms

We here provide the full Peano axiomatization of each domain.

## D.1 Arithmetic

Our axiomatization of arithmetic includes the classical unary definition of the natural numbers (zero and successor function), axioms about addition, multiplication, and the principle of induction.

```
= : [('t : type) -> 't -> 't -> prop].

nat : type.

z : nat.
s : [nat -> nat].

+ : [nat -> nat -> nat].
* : [nat -> nat -> nat].

+_z : [('n : nat) -> (= (+ 'n z) 'n)].
+_s : [('n : nat) -> ('m : nat) -> (= (+ 'n (s 'm)) (s (+ 'n 'm)))].

*_z : [('n : nat) -> (= (* 'n z) z)].
*_s : [('n : nat) -> ('m : nat) -> (= (* 'n (s 'm)) (+ 'n (* 'n 'm)))].

nat_ind : [('p : [nat -> prop]) -> ('p z) -> [('n : nat) ->
            ('p 'n) -> ('p (s 'n))] -> [('n : nat) -> ('p 'n)]].

#backward nat_ind.
#forward +_z ((+ 'n z) : nat).
#forward +_s ((+ 'n (s 'm)) : nat).
#forward *_z ((* 'n z) : nat).
#forward *_s ((* 'n (s 'm)) : nat).
```

## D.2 Propositional Logic

```
prop : type.

false : prop.

/* Connectives */
not : [prop -> prop].
```

```
and : [prop -> prop -> prop].
or : [prop -> prop -> prop].
iff : [prop -> prop -> prop].

/* Introduction rule for conjunction */
#backward and_i.
and_i : [('P : prop) -> ('Q : prop) -> 'P -> 'Q -> (and 'P 'Q)].
/* Elimination rules for conjunction */
#forward and_el ('_ : (and 'P 'Q)).
and_el : [('P : prop) -> ('Q : prop) -> (and 'P 'Q) -> 'P].
#forward and_er ('_ : (and 'P 'Q)).
and_er : [('P : prop) -> ('Q : prop) -> (and 'P 'Q) -> 'Q].

/* Introduction rules for disjunction */
#backward or_il.
or_il : [('P : prop) -> ('Q : prop) -> 'P -> (or 'P 'Q)].
#backward or_ir.
or_ir : [('P : prop) -> ('Q : prop) -> 'Q -> (or 'P 'Q)].
/* Elimination rule for disjunction */
#backward or_e infer infer infer infer subgoal subgoal.
or_e : [('P : prop) -> ('Q : prop) -> ('R : prop) ->
        (or 'P 'Q) -> ['P -> 'R] -> ['Q -> 'R] -> 'R].

/* Introduction rule for negation */
#backward not_i.
not_i : [('P : prop) -> ['P -> false] -> (not 'P)].
/* Elimination rule for negation */
not_e : [('P : prop) -> (not 'P) -> 'P -> false].
#backward exfalso.
exfalso : [false -> ('P : prop) -> 'P].

/* Introduction rules for equivalence */
#backward iff_i.
iff_i : [('P : prop) -> ('Q : prop) -> ['P -> 'Q] -> ['Q -> 'P] -> (iff 'P 'Q)].
/* Elimination rules for equivalence */
#forward iff_el ('_ : (iff 'P 'Q)).
iff_el : [('P : prop) -> ('Q : prop) -> (iff 'P 'Q) -> ['P -> 'Q]].
#forward iff_er ('_ : (iff 'P 'Q)).
iff_er : [('P : prop) -> ('Q : prop) -> (iff 'P 'Q) -> ['Q -> 'P]].

/* Excluded middle */
#forward em.
em : [('P : prop) -> (or 'P (not 'P))].
```

### D.3  Groups

```
= : [('t : type) -> 't -> 't -> prop].

G : type.

op : [G -> G -> G].
id : G.

/* Associativity */
#forward op_assoc ((op (op 'a 'b) 'c) : G).
op_assoc : [('a : G) -> ('b : G) -> ('c : G) -> (= (op (op 'a 'b) 'c) (op 'a (op 'b 'c)))].

/* Commutativity */
#forward op_comm ((op 'a 'b) : G).
```

```
op_comm : [('a : G) -> ('b : G) -> (= (op 'a 'b) (op 'b 'a))].

/* Identity */
#forward id_l.
id_l : [('a : G) -> (= (op id 'a) 'a)].

/* Inverse */
inv : [G -> G].
#forward inv_l.
inv_l : [('a : G) -> (= (op (inv 'a) 'a) id)].
```

# E   Extrinsic evaluation problems

Here we list all problems used in the extrinsic evaluation in Arithmetic and Propositional Logic.

## E.1   Arithmetic

The following are the 10 problems we extracted from from the Natural Number Game that (a) don't use previous lemmas, being reasonably provable straight from the axioms, (b) are about natural numbers (so, for example, this excludes the Proposition World, which is essentially about propositional logic). The prefix in these problems tell which "world" of the game it came from: t stands for the Tutorial World, a is for the Addition World, and finally m is for the Multiplication World.

```
t_example1. [('x : nat) -> ('y : nat) -> ('z : nat) -> (= (+ (* x y) z) (+ (* x y) z))]
t_example2. [('x : nat) -> ('y : nat) -> (= 'y (+ 'x n7)) -> (= (* n2 'y) (* n2 (+ 'x n7)))]
t_example3. [('a : nat) -> ('b : nat) -> (= (s 'a) 'b) -> (= (s (s 'a)) (s 'b))]
t_add_succ_zero. [('a : nat) -> (= (+ 'a (s z)) (s 'a))]
a_zero_add. [('n : nat) -> (= (+ z 'n) 'n)]
a_add_assoc. [('a : nat) -> ('b : nat) -> ('c : nat) -> (= (+ (+ 'a 'b) 'c) (+ 'a (+ 'b 'c)))]
a_succ_add. [('a : nat) -> ('b : nat) -> (= (+ (s 'a) 'b) (s (+ 'a 'b)))]
a_succ_eq_add_one. [('n : nat) -> (= (s 'n) (+ 'n (s z)))]
m_zero_mul. [('m : nat) -> (= (* z 'm) z)]
m_mul_one. [('m : nat) -> (= (* 'm (s z)) 'm)]
```

Notably, we give below the full proof our best agent for Arithmetic finds for a_succ_add:

```
theorem a_succ_add : [('a0 : nat) -> ('a1 : nat) -> (= (+ (s 'a0) 'a1) (s (+ 'a0 'a1)))] {
    intro x : nat.
    apply nat_ind.
    goal (= (+ (s x) z) (s (+ x z))) {
        show (= (+ (s x) z) (s x)) by +_z.
        show (= (+ x z) x) by +_z.
        show (= x (+ x z)) by eq_symm.
        show (= (+ (s x) z) (s (+ x z))) by rewrite.
    }
    goal [('n : nat) -> (= (+ (s x) 'n) (s (+ x 'n))) ->
          (= (+ (s x) (s 'n)) (s (+ x (s 'n))))] {
        intro x0 : nat.
        intro x1 : (= (+ (s x) x0) (s (+ x x0))).
        show (= (s (+ x x0)) (+ (s x) x0)) by eq_symm.
        show (= (+ (s x) x0) (+ (s x) x0)) by rewrite.
        show (= (s (+ x x0)) (s (+ x x0))) by rewrite.
        show (= (+ x (s x0)) (s (+ x x0))) by +_s.
        show (= (+ x (s x0)) (+ (s x) x0)) by rewrite.
        show (= (s (+ x x0)) (+ x (s x0))) by eq_symm.
        show (= (+ (s x) x0) (+ x (s x0))) by rewrite.
        show (= (+ x (s x0)) (+ x (s x0))) by rewrite.
        show (= (+ (s x) (s x0)) (s (+ (s x) x0))) by +_s.
```

```
        show (= (+ (s x) (s x0)) (s (+ x (s x0)))) by rewrite.
    }
}
```

This proof is not minimal – our human-written Peano proof for this level of the game has 7 steps in the inductive case, compared to 12 in the proof the agent finds. Nevertheless, this is a non-trivial theorem to prove by hand in this low-level axiomatic system, and our agent was not explicitly trained to target this theorem. Thus, this shows that it does learn general patterns that are useful in proof search.

### E.2  Propositional Logic

We here list all the statements in Theorem 41 of Kleene's book [20], "Introduction to Metamathematics", represented in Peano:

```
1.  [('A : prop) -> ['A -> 'A]]
2.  [('A : prop) -> ('B : prop) -> ('C : prop) -> ['A -> 'B] -> ['B -> 'C] -> ['A -> 'C]]
3.  [('A : prop) -> ('B : prop) -> ('C : prop) -> ['A -> ['B -> 'C]] -> ['B -> ['A -> 'C]]]
4.  [('A : prop) -> ('B : prop) -> ('C : prop) -> ['A -> ['B -> 'C]] -> [(and 'A 'B) -> 'C]]
5.  [('A : prop) -> ('B : prop) -> ('C : prop) -> [(and 'A 'B) -> 'C] -> ['A -> ['B -> 'C]]]
6.  [('A : prop) -> ('B : prop) -> ('C : prop) -> ['A -> 'B] -> [['B -> 'C] -> ['A -> 'C]]]
7.  [('A : prop) -> ('B : prop) -> ('C : prop) -> ['A -> 'B] -> [['C -> 'A] -> ['C -> 'B]]]
8a. [('A : prop) -> ('B : prop) -> ('C : prop) -> ['A -> 'B] -> [(and 'A 'C) -> (and 'B 'C)]]
8b. [('A : prop) -> ('B : prop) -> ('C : prop) -> ['A -> 'B] -> [(and 'C 'A) -> (and 'C 'B)]]
9a. [('A : prop) -> ('B : prop) -> ('C : prop) -> ['A -> 'B] -> [(or 'A 'C) -> (or 'B 'C)]]
9b. [('A : prop) -> ('B : prop) -> ('C : prop) -> ['A -> 'B] -> [(or 'A 'C) -> (or 'B 'C)]]
10a. [('A : prop) -> ('B : prop) -> [(not 'A) -> ['A -> 'B]]]
10b. [('A : prop) -> ('B : prop) -> ['A -> [(not 'A) -> 'B]]]
11. [('A : prop) -> ('B : prop) -> ['B -> ['A -> 'B]]]
12. [('A : prop) -> ('B : prop) -> ['A -> 'B] -> [(not 'B) -> (not 'A)]]
13. [('A : prop) -> ('B : prop) -> ['A -> (not 'B)] -> ['B -> (not 'A)]]
14. [('A : prop) -> ('B : prop) -> [(not 'A) -> 'B] -> [(not 'B) -> 'A]]
15. [('A : prop) -> ('B : prop) -> [(not 'A) -> (not 'B)] -> ['B -> 'A]]
16. [('A : prop) -> ('B : prop) -> ['A -> 'B] -> ['B -> 'A] -> (iff 'A 'B)]
17a. [('A : prop) -> ('B : prop) -> (iff 'A 'B) -> ['A -> 'B]]
17b. [('A : prop) -> ('B : prop) -> (iff 'A 'B) -> ['B -> 'A]]
18a. [('A : prop) -> ('B : prop) -> (iff 'A 'B) -> 'A -> 'B]
18b. [('A : prop) -> ('B : prop) -> (iff 'A 'B) -> 'B -> 'A]
19. [('A : prop) -> (iff 'A 'A)]
20. [('A : prop) -> ('B : prop) -> (iff 'A 'B) -> (iff 'B 'A)]
21. [('A : prop) -> ('B : prop) -> ('C : prop) -> (iff 'A 'B) -> (iff 'B 'C) -> (iff 'A 'C)]
22. [('A : prop) -> ('B : prop) -> ('C : prop) -> ['A -> ['B -> 'C]] ->
     [(not (not 'A)) -> [(not (not 'B)) -> (not (not 'C))]]]
23. [('A : prop) -> ('B : prop) -> [(not (not ['A -> 'B]))] ->
     [(not (not 'A)) -> (not (not 'B))]]
24. [('A : prop) -> ('B : prop) -> ('C : prop) -> [(not (not ['A -> 'B]))] ->
     [(not (not ['B -> 'C]))] -> [(not (not ['A -> 'C]))]]
25. [('A : prop) -> ('B : prop) ->
     (iff (not (not (and 'A 'B))) (and (not (not 'A)) (not (not 'B))))]
```

We use the numbering from the book here — there are 30 statements, even if the last one is labeled 25.

## F  Discussion: extension to other proof assistants

While our approach is implemented in Peano, it can in principle be extended to mainstream theorem provers like Lean and Coq. Peano's typing rules (a version of the Calculus of Constructions, without native inductive types) are essentially a subset of these systems' more expressive calculi, and its minimalism makes it particularly suitable for automated proof search. The most practical path to

integration would be similar to recent tools like Duper [9]: embedding problems from the target system (e.g., Lean) into a Peano representation, running the prover, and then reconstructing the discovered proofs in the original system. While developing a Peano-to-Lean proof object translator would be relatively straightforward given the simplicity of Peano's type system, translating arbitrary Lean problems to Peano would require significant more engineering effort (e.g., we would need to generate explicit axioms for inductive types and quotients, which are not part of Peano). However, we believe a simplified environment is valuable for investigating how self-improving agents can build towards deeper theorems, eventually being able to invent new definitions and accumulate a growing library (which MINIMO does not, yet). Once these capabilities are demonstrated convincingly, the engineering investment in proof translation will become more compelling.

