# OpenReview forum: "Learning Formal Mathematics From Intrinsic Motivation"
_NeurIPS.cc/2024/Conference — NeurIPS 2024 oral_

### Official Review · Reviewer_awrR · 2024-07-01

**Soundness:** 1
**Presentation:** 2
**Contribution:** 3
**Rating:** 6
**Confidence:** 4

**Summary:**

This work targets the problem of formal theorem-proving, in the particular setting where a learning agent starts only from axioms (contrary to most other works which use models trained on a plethora of mathematics data already) similar to an AlphaZero style setup. They use a single neural model (transformer) to learn to (1) produce conjectures (2) a distribution over proof steps (3) a value function for proof search. In an alternating process, conjectures are sampled from the model using constrained decoding, on which proof search is then attempted, yielding data for further training, with the goal of producing a model capable of generating harder conjectures as its proof-writing ability increases. They also incorporate hindsight relabeling in this setting. The results indicate that the model does learn to generate harder conjectures and become better at proving them. On a set of known theorems from arithmetic and propositional logic, the model is capable of improving as it learns to prove more of the conjectures it generates.

**Strengths:**

The paper discusses a question of great significant in the field of AI-for-mathematics: How can we produce systems which can learn from scratch (from axioms)? The topics of conjecturing and proving have, as far as I know, not been heavily investigated $\emph{together}$ and this work serves as a nice exploratory piece on this area, which I suspect will become more popular. While there are prior works on RL for theorem proving, the use of intrinsic motivation from RL in this field is a first, to the best of my knowledge.

**Weaknesses:**

1. Lack of evidence: While conjecturing is a main component of the approach, examples of conjectures provided by the model (at any stage) are not provided neither in the main content nor in the appendix. It is hard to judge the quality of such conjectures or to empirically see an improvement in conjecture difficulty over phases when none are provided.

Similarly, only a single proof is included in the entire submission, in the appendix. It is useful for the reader to get a sense of what kinds of proofs the model is finding, especially amongst each of the three domains, on both extrinsically defined goals and those the conjecturer comes up with.

Additionally, only the propositional logic and arithmetic tasks have an extrinsically defined set of theorems for testing. Why is a similar set not included for the abelian group task? Judging from Figures 2 and 3, the learning dynamics for the group theory task are different than the other two tasks.

2. Conjecturing & intrinsic motivation: It is unclear how much the agent itself is producing conjectures that it is intrinsically motivated to solve. As I understand it, the conjectures are generated via a constrained decoding on the language model, but I'm not sure how much the LM would actually generate conjectures without the constraint after being trained for a while. How much is the conjecturing really improving, given that at the end of each proof search round, only 10% of conjectures are considered hard, 40% easy, and the rest trivial. Perhaps it is better to discretize difficulty based on proof length?

3. Poor scalability: Even with the modest compute resources as described in the appendix, the tasks are quite simple, as are the extrinsically defined goals, which the model shows improvement on mostly on the arithmetic task sourced from NNG. On the propositional logic task, the model at the end of the fifth policy can only solve 4(?) more of 30 problems as compared to the randomly initialized model from the 0th iteration on the policy.

Some typos I noticed:
1. Line 169: "Mathematically, $\mathcal{C}$ is a function $f_{\mathcal{C} : \Sigma* \to $..." but $\mathcal{C}$ is a set.
2. Line 245: "levarage" -> leverage
3. Figure 2 caption: I think you meant "evaluated"
4. Line 341: "recevies" -> "receives"
5. Line 345: "training tends to collapse to ? by"

**Questions:**

1. Given the simple evaluation domains, the lengths of proofs for conjectures (measured in # of tactics) at the end of each iteration could be a useful indicator for how the both how the proof search improves over time, and how difficult the conjectures become. Can you report numbers measuring problem difficulty as length of proofs? Can you include examples of proofs found over iterations?

2. How often does the conjecturer produce "trivial" conjectures? Do you have any measure as to how sound the completion engine is as iterations proceed? I imagine that the first round of conjectures is entirely random given the model is untrained. Can you include examples of conjectures produced over the iterations?

3. It seems unclear whether this sort of learning in multiple phases is better than just performing one single phase. In AlphaGeometry (https://www.nature.com/articles/s41586-023-06747-5), though geometry is a particularly pointed domain, all data is generated upfront and then a model is trained. In this paper, the extra training data produced from hindsight relabeling is crucial, they indicate on line 336 that the approach does not reach its intended goal without it. Perhaps upfront training may be better? It could be the case that by placing a somewhat more complex distribution on the constrained decoding, one can generate many harder yet valid conjectures without learning a transformer first. Can the authors comment on this?

4. On page 22, Is the supplied full proof for a_succ_add exactly that found by the proof search? For example, is the step "show (= x (+ x z)) by eq_symm" predicted by the model, or just "eq_symm"? Similarly does the model produce "intro x1 : (= (+ (s x) x0) (s (+ x x0)))" or just "intro x1".
If the case of the former, does the Peano language automatically enumerate all relevant types and reachable hypotheses? Otherwise generating a type like "(= (+ (s x) x0) (s (+ x x0))" might make the action space infinite?

5. The performance of the last checkpoint compared to the initial checkpoint on the propositional logic extrinsic set is not very different, can you comment on why this might be the case? That extrinsic set, as specified on page 23, does not seem to be particularly challenging?

6. Sampling conjectures from a small model should be fairly quick. Can the authors comment on how much time is spent generating conjectures and proving conjectures separately? Only the aggregated number is reported in the appendix section A.

**Limitations:**

I believe the authors have adequately addressed the limitations and broader societal impacts of their work.

---

> ### Author Rebuttal · Authors · 2024-08-07
>
> We thank the reviewer for the through assessment of our work, and the comment on the significance of our research question! We provide responses below, and additional examples in the global response.
>
> > Can you report numbers measuring problem difficulty as length of proofs? Can you include examples of proofs found over iterations?
>
> Please refer to the global response for numbers on the proof length per iteration. We also included a few examples of proofs with the lowest log-probs found during training (proofs found on average get longer).
>
> > How often does the conjecturer produce "trivial" conjectures? Do you have any measure as to how sound the completion engine is as iterations proceed? I imagine that the first round of conjectures is entirely random given the model is untrained. Can you include examples of conjectures produced over the iterations?
>
> At first, most of the conjectures that are successfully proved are trivial (we analyzed provability in Appendix C), and many are trivially false (e.g. '!!false' in propositional logic). Hindsight replay allows the agent to uncover non-trivial conjectures and learn. We refer to the global response for concrete examples of conjectures.
>
> > It seems unclear whether this sort of learning in multiple phases is better than just performing one single phase, like in AlphaGeometry. [...] Perhaps upfront training may be better? Can the authors comment on this?
>
> Good question! Earlier in the project, we did consider trying something like the AlphaGeometry approach. There are two main challenges for this. One is that, in AlphaGeometry, problems and solutions are simultaneously generated in the forward direction only, both simultaneously, by creating new hypotheses and applying forward deduction. While that works for the Euclidean geometry axioms, it does not generalize easily to a general formal system like dependent type theory. As an example of where it becomes hard, in arithmetic, our conjecturer generates several statements that are then proved by induction (as it is likely to generate something equivalent to forall ('n : nat), …). For these conjectures to be produced in the forward direction, like in AlphaGeometry, we would need to independently stumble upon proofs of matching base and inductive cases without actually planning to do so. It's also not entirely clear how the proof of the inductive case would be generated in the forward direction, without having a goal. The other challenge is that AlphaGeometry also relied on an existing, complete, domain-specific deductive closure solver for Geometry to generate data, which our approach for data generation doesn't require (and which doesn't exist for dependent type theory, given that the deductive closure can be infinite).
>
> > On page 22, Is the supplied full proof for a_succ_add exactly that found by the proof search? [...] does the Peano language automatically enumerate all relevant types and reachable hypotheses?
>
> We reconstruct the proof in Peano concrete syntax from the proof search tree as it is found (to get the equivalent proof as if human-written), though at each step the model gets the proof state, current goal, and only chooses the next step. But you are correct, Peano enumerates the relevant types and reachable hypotheses. So for 'intro', which takes the next universally quantified variable from the goal into the state, it suffices for the model to choose 'intro' as an action, as the rest is forced (both the name - we just generate a numbered identifier, and its type is taken from the goal by Peano). For 'eq_symm', eq_symm can be applied to any equality currently in the state, so Peano will enumerate those (not with a particular algorithm for eq_symm, just looking at the type signature of eq_symm). So in this case the policy first chooses 'eq_symm', then chooses one of the valid results to get from it in the current state.
>
> > The performance of the last checkpoint compared to the initial checkpoint on the propositional logic extrinsic set is not very different, can you comment on why this might be the case?
>
> Given that our agent is only trained on its own self-generated problems, there is no a priori guarantee that it gets better at solving human-written problems, which are only a very small subset of what is formally true (one open problem is how to characterize this subset that we find interesting, to focus discovery on it). Looking at the conjectures generated and proved by our agent, it is clear that they do not generally resemble human theorems. (see examples in the global response)
>
> Thus, our extrinsic set served just to give a signal of whether there is some intersection between where the improvements take place and the set of problems that are interesting to humans. We observe some improvement, although it is still small in absolute terms, as you point out. We think a really interesting and important problem that this suggests is exactly how to structure the learning process so that the conjectures evolve in better alignment with theorems that are interesting for humans. Our setup can serve as a starting point for this investigation.
>
> > Sampling conjectures from a small model should be fairly quick. Can the authors comment on how much time is spent generating conjectures and proving conjectures separately? Only the aggregated number is reported in the appendix section A.
>
> You are correct, conjecture generation in each iteration is fairly quick. Sampling 200 new, unique conjectures takes 1-3 minutes in each iteration, whereas proof search on 200 conjectures takes 2-4 hours of compute (which our implementation distributes across several prover processes, since proof attempts in each batch are independent). Sampling is mostly GPU-bound, while proof search also spends measurable time on Peano enumerating actions during MCTS, especially in deep states where the state can get large.
>
> Thank you again for the detailed review. We're happy to engage further if any other questions remain!

---

> > ### Comment · Reviewer_awrR · 2024-08-12
> >
> > Thanks for the detailed response, which has addressed most of my questions.
> >
> > > Given that our agent is only trained on its own self-generated problems, there is no a priori guarantee that it gets better at solving human-written problems, which are only a very small subset of what is formally true
> >
> > While I think this piece is a good initial exploration into this area, ultimately I would feel that the goal should be either to (1) produce systems capable of solving problems interesting to humans or (2) doing "alien" mathematics. I feel that the results are not particularly enlightening either way. I think it will take more than straightforward modifications of the learning dynamics to achieve success on, for example, this held-out set of human problems in the arithmetic task. Especially since the compute requirements are quite high to get to the current results.
> >
> > What do you see as a way to structure the learning process to be able to prove all of these held-out theorems. For example, will more training rounds suffice?

---

> > > ### Author Response · Authors · 2024-08-13
> > >
> > > We thank the reviewer for reading through our previous response! We're glad it addressed most of your questions.
> > >
> > > > While I think this piece is a good initial exploration into this area, ultimately I would feel that the goal should be either to (1) produce systems capable of solving problems interesting to humans or (2) doing "alien" mathematics.
> > >
> > > We fully agree that either (1) or (2) would be the most interesting ultimate goals. We would argue that our system is closer right now to "doing alien mathematics", in the sense that it does self-improve and prove thousands of theorems that are not given to it, though these aren't yet of special interest to humans. Doing "deep" alien mathematics (in the sense of developing full mathematical theories about novel mathematical objects) will be possible with library learning - growing a library of lemmas and new definitions. Still, in either case, we would want to understand what it takes to make such alien mathematics "interesting" (aligning or not with human mathematics). We completely concede that we don't close these fundamental questions with this work, but rather open them up for study in a concrete setting with the setup we propose.
> > >
> > > > Especially since the compute requirements are quite high to get to the current results.
> > >
> > > Our compute requirements are modest compared to prior work that attempts self-improvement in formal mathematics. Each of our runs could be completed in a day on a single academic node, as mentioned in Appendix A. In contrast, expert iteration in prior work (typically in Lean) required a much larger scale to see improvements. For instance, in gpt-f lean [1], each run took "about 2000 A100 days of compute"; this leads to solving 10 extra problems in minif2f. HTPS [2] employed 232 A100s for 7 days for each of their runs in Lean; the improvement from day 2 to day 7 was 29 extra training problems (minif2f-curriculum). Given the small absolute scale of those gains, it seems unlikely that they would be able to measure improvements at all within a day on a single GPU, as we do (> 1000x less compute). Our results could certainly be improved by increasing scale, as our setup is also embarrassingly parallel, but we believe it already allows interesting deep questions to be studied with much less compute.
> > >
> > > [1] Polu, Stanislas, et al. "Formal mathematics statement curriculum learning." arXiv preprint arXiv:2202.01344 (2022).
> > > [2] Lample, Guillaume, et al. "Hypertree proof search for neural theorem proving." NeurIPS 2022
> > >
> > > > What do you see as a way to structure the learning process to be able to prove all of these held-out theorems. For example, will more training rounds suffice?
> > >
> > > This is a really interesting question! While a larger scale (e.g. more rounds, larger batches per round, larger Transformer, more search) could all likely improve the extrinsic results, it would not yet be the most interesting. We think that studying what to reward during self-improvement that maximizes performance in held-out problems will first help make the "scaling law" (improvement x compute) more favorable, for scaling up to then make sense. Here is a notable observation and ideas coming from analyzing our existing runs:
> > >
> > > Human mathematicians tend to prefer more general theorems. Our agent doesn't. In fact, the conjecturer often ends up adding unnecessary/unrelated assumptions to the theorem statement as a way to make conjectures harder for the prover, because those add more actions to the action space and thus make search more challenging. There are two possible ideas to mitigate this:
> > > * For the conjecturer, we can reward it when it produces conjectures that generalize previous results (there are simple symbolic strategies for checking whether a theorem A trivially follows from theorem B, by checking whether A is a substitution instance of B), or penalize it when it produces conjectures that trivially follow from previous ones (using the same check).
> > > * For the prover, one could try a related data augmentation strategy, where after proving A, we can synthesize theorems that consist of A with extra assumptions (thus, essentially the same proof still works, after doing 'intro' on the unnecessary assumptions). Training on these would help the prover not find theorems with unnecessary assumptions to be harder, and thus the conjecturer would be discouraged from generating those.
> > >
> > > We don't know the effect of either of these strategies yet, but this is the kind of investigation that our setup allows future work to explore. Besides, the RL literature on intrinsic rewards is very rich, and we only scratch the connection here. We will include a discussion of these directions in the paper, emphasizing that should they work here, they can also translate to novel domains (with no pre-existing data).
> > >
> > > We thank you again for the thoughtful response. Please let us know if any other questions remain, and we'd be happy to discuss them before the discussion period ends.

---

> ### Comment · Reviewer_awrR · 2024-08-13
>
> I thank the authors for the interesting remarks regarding my question, and the comments about compute requirements. I feel that works like GPT-f and HTPS are on the extreme end of the compute scale, and there are many reasonable works exist that show "good results" with a fairer compute budget.
>
> The primary concern I have is how to judge an adequate attempt at a highly interesting problem. The results are not particularly convincing, especially when referring to performance on the extrinsic set of problems, and the conjectures generated are closer to "alien math" (which is fine) but also seem to not convey any particularly interesting results, even on a simple level. However, this paper is the first to attempt this conjecturing-proving setup as far as I know. I think the particularly interesting research ideas would be those which show real improvements on this task. Further consideration of this is better done by the metareviewer.
>
> Because the authors have nicely addressed my questions and clarified their work, I have increased my score.

---

### Official Review · Reviewer_XdUe · 2024-07-10

**Soundness:** 2
**Presentation:** 4
**Contribution:** 3
**Rating:** 6
**Confidence:** 3

**Summary:**

The authors propose a novel approach to formal theorem proving (FTP) that leverages a language model's self-improvement capabilities by framing mathematical theorem proving as a sequential game involving generating a conjecture to be proven, and then proving it, and so forth. The approach consists of two primary steps: first, the authors use a constrained decoding approach to generate mathematically valid conjectures of a target difficulty, treating an arbitrary language model (LM) as a probability distribution over conjectures in a given domain. Then, the generated conjectures are solved using an MCTS-guided policy, which allows for backtracking-based relabeling for improved reward shaping signal during policy learning. Experimental results conducted in the Peano language on three different domains of theorems show that the proposed approach can find increasingly challenging conjectures and train a policy that solves them, and that this learned policy can also solve human-written conjectures of interest.

**Strengths:**

* Self-guided improvement of LLMs is a powerful paradigm that has shown novel improvements in a number of challenging areas, such as code generation and prose writing. Approaching the important problem of automated theorem proving from a lens of self-improvement is novel and valuable and I believe can provide a foundation for a new class of approaches to LLM-guided FTP.
* The game-theoretic framing of the problem is intuitive and MCTS, under the structure that the Peano environment provides, is a smart way to frame policy learning that makes hindsight relabeling efficient. The usage of hindsight relabeling to further guide conjecture generation and policy training is valuable.
* The paper is very well written and generally easy to follow.

**Weaknesses:**

* There is a substantial body of work on the self-improvement of LLMs that I felt was not adequately addressed by the authors. Although I am not familiar with an existing self-improvement approach in the space of FTP specifically, there are numerous works like [1] that allow LLM-generated content to improve themselves. The paradigm proposed in this paper is not exactly the same, but a comparison to and acknowledgement of this related work should be included. The 'mathematical conjecturing' section of the related work could be condensed (or moved to the appendix) in favor of covering this topic that appears to have more recent and relevant literature.
* The MCTS policy and learning approach was a bit light on detail. What exactly is the state for the MCTS policy? What are the details of each operator in the MCTS approach? This information will help the reader understand just how general the approach proposed is for general FTP and not just Peano-based domains. This could be included even in the appendix.
* To the previous point - it would be nice to see the authors include a more thorough discussion regarding if and how their approach could extend to popular theorem proving environments like Coq and Lean. Is it applicable? Why or why not? The strongest addition would be to actually show the approach in one of these environments. I don't think it's necessarily a knock on the approach to be instantiated only in Peano, but the community would greatly benefit from an instantiation of this approach in Coq or Lean, or at least a discussion of how and why this can be done.
* The idea of using the (fixed) learned policy itself to evaluate 'difficulty' of conjectures does not totally inspire confidence in the claims made in the paper. Is there a qualitative analysis for RQ1 that can be done on these conjectures to show that they indeed become harder over time?

Overall, I think the contribution is novel and of great interest to the community, but I am left unsure about a lot of details regarding the approach and its design decisions. I am happy to increase my score based on the authors' response.

[1] Language Models Can Teach Themselves to Program Better. Haluptzok et. al. ICLR 2023.

**Questions:**

* Later iterations of the policy seem to be a bit worse on (or consider a bit harder) "easy" conjectures, and seem to do better on (or consider to be easier) "harder" conjectures (as evidenced by Fig. 2.)  Why do these further trained policies have more trouble discerning what is "easy" and what is "hard"? Is it just generally 'good' at solving all conjectures? Or would it be useful to have a curriculum-style approach that occasionally asks "easy" conjectures even later in training?
* Why do we start with a randomly initialized Language Model? How does the choice of Language Model affect the performance of conjecture generation?
* I didn't seem to fully understand how the approach chooses a specific difficulty of conjecture generator. I understand where difficulty 'scores' come from, but how do we choose a conjecture based on its difficulty?
* Can the authors provide explanation for if/how to apply the approach to environments like Coq and Lean?

**Limitations:**

Limitations are discussed in the paper but there are some questions about the limitations of the approach and its applicability to other environments (that I have detailed above.)

---

> ### Author Rebuttal · Authors · 2024-08-07
>
> Thank you for the through review and encouraging comments! We agree that several of the clarifications you requested should be addressed in the paper, and will do so (including the additional examples in the global response).
>
> > There is a substantial body of work on the self-improvement of LLMs that I felt was not adequately addressed by the authors.
>
> We agree. Some prior work has also focused on self-improvement on informal math (eg GSM8k), which is also relevant, though none that we know of do it for FTP or without pre-training data. We will acknowledge this literature and add references accordingly.
>
> > MCTS was a bit light on detail. What exactly is the state for the MCTS policy? [...]
>
> This makes sense, we will include concrete examples of states and MCTS expansions in the Appendix. The state in Peano has the same form as the proof state in the interactive modes of Coq and Lean – a set of objects with their types, and a goal type. We just represent it as a string to feed to the Transformer model, so nothing Peano-specific in what the LLM does. One random example:
>
> "State: { x : G, x0 : G }\nGoal: [('a2 : G) -> (= (op id 'a2) 'a2)]"
> (the state has two elements of the group G and still has a 'forall a2 : G, id * a2 = a2' in the goal)
>
> The proof search method we use is broadly similar to the instantiations of MCTS is used in recent prior work like in gpt-f or Holophrasm. The main difference is that, since Peano has a finite action space, we don't sample the action as a sequence of tokens from the policy, but rather just score the enumerated actions using the LM. Other search algorithms, like HTPS, can well be used in our setup as well instead.
>
> > To the previous point - [...] discussion regarding if and how their approach could extend to popular theorem proving environments like Coq and Lean
>
> Yes, good question! Peano's typing rules are essentially a subset of Lean's and Coq's (CoC vs CIC, with variations), and its simplicity makes it more appropriate to do proof search in it. It would be relatively simple to develop a Peano -> Lean proof object translator, that takes proof objects found in Peano and generates the equivalent Lean proof object. The opposite direction would take more engineering work, but is also possible (the additional complexity of the Lean type system makes it much more convenient for users, but is not fundamentally required to represent mathematics). Thus, to make our approach available in Lean (for instance), the shortest path is likely to be something like the recent Duper (https://github.com/leanprover-community/duper), which takes the problem in Lean, translates it to the external prover, calls the prover, and then attempts to reconstruct the proof in Lean. We think Peano's more minimal type system makes it suitable to investigate how to get the self-improving agent to eventually build towards much deeper theorems (making new definitions and a library on the way), and at the point where that works it will pay off to spend the engineering effort to build proof object translators.
>
> > Is there a qualitative analysis for RQ1 that can be done on these conjectures to show that they indeed become harder over time?
>
> Please refer to the global response for examples of conjectures that show up during training. We will include those (for all 3 domains) in the Appendix. Overall, we see both the longest proofs that the agent is capable of finding getting longer, as well as the hardest conjectures being more complex. Since the agent still doesn't make new definitions or reuse past lemmas, the conjectures however don't yet evolve in complexity in a human-like way, which we believe to be an extremely interesting direction for future work that our setup can be used to explore.
>
> > Later iterations of the policy seem to be a bit worse on (or consider a bit harder) "easy" conjectures
>
> That's true, we do seem to observe a distribution drift over time in terms of conjectures. As the conjecturer starts to focus on hard ones (which in this case, since the agent doesn't make up new definitions to shorten higher-level statements, tends to mean longer conjectures), the policy seems to be slightly more uncertain in easier problems than the policy at iterations 1-2 (which have been trained more recently on easier conjectures). Yes, we believe a curriculum-style approach could be useful to mitigate this behavior. However, in terms of proof search, this uncertainty tends to not be that much of a problem in easier conjectures because they have either shorter proofs or lower branching factor (thus, MCTS still finds the proof even if the agent prioritizes other branches at first).
>
> > Why do we start with a randomly initialized LM? How does the choice of LM affect performance?
>
> Our main goal is to build towards a self-improving pipeline that does not require pre-training data, much like AlphaZero did for board games. If this is achieved, then our agent will be able to produce mathematics in both existing domains for which we have plentiful training data, and for novel domains for which we do not. For the LM, we chose a standard GPT-2-based Transformer architecture, but we don't believe this is crucial to the idea, as work in LLMs generally show that scale and data matter more than architecture details (sticking to Transformers specifically).
>
> > [...] how the approach chooses a specific difficulty of conjecture generator.
>
> We always try to generate hard conjectures. Here, "try" means that we condition the Transformer on the 'hard' difficulty. After proof search on a batch of samples, we get the actual observed difficulty by whether the prover succeeded and, if so, the log-probability of the proof under the current LM. That feedback signal is then used to further train the conjecturer (by generating LM training examples where the conjecture's statement is paired with the observed difficulty, rather than always 'hard').
>
> Thanks again for the review. We're happy to engage or clarify further!

---

> > ### Comment · Reviewer_XdUe · 2024-08-09
> > **Thanks for your response**
> >
> > Thanks to the authors for the engaging and detailed response!
> >
> > >We think Peano's more minimal type system makes it suitable to investigate how to get the self-improving agent to eventually build towards much deeper theorems (making new definitions and a library on the way), and at the point where that works it will pay off to spend the engineering effort to build proof object translators.
> >
> > I think a discussion of this would be of great use to include in the paper. This is probably more appendix material than main text, but readers will definitely want to know what advantages Peano brings in this specific context and how it could feasibly be extended to existing theorem provers. I highly encourage the authors to include an even more thorough version of this discussion in the paper.
> >
> > The additional qualitative results are also useful and I hope the authors include these and a few more in the main text, along with a comprehensive analysis of them (which includes the metrics recorded in the general response.)
> >
> > Conditioned on these points, I am increasing my score. I look forward to the extended discussion that the authors will be adding to the main text.

---

> > > ### Author Response · Authors · 2024-08-13
> > >
> > > We sincerely thank the reviewer for engaging with our work and the response! We agree that this discussion on how this setup can connect to existing theorem provers is extremely important for readers. We will expand and include this discussion in the paper.
> > > We will also include the examples and descriptions from the previous discussion to improve on the original points of confusion you had raised. Thank you again!

---

### Official Review · Reviewer_6DqB · 2024-07-12

**Soundness:** 3
**Presentation:** 3
**Contribution:** 3
**Rating:** 6
**Confidence:** 3

**Summary:**

This paper proposes to create mathematical agents by jointly learning to posing challenging problems (conjecturing) and solving the problems (theorem proving). Specifically, they use a randomly initialized Transformer to perform both conjecturing and proof search in a loop.

**Strengths:**

The proposed method is demonstrated generating more difficult conjectures and obtaining better theorem proving policy over several iterations. Experiments are also conducted on proving human-written theorems.

**Weaknesses:**

1. Further justification of using log-likelihood as the main evaluation in this paper.
2. How does this method facilitates solving dataset such as miniF2F and mathlib?

**Questions:**

Q1: How to guarantee f_C soundness and completeness?
Q2: In Figure 2, propositional logic, the search baseline policy at iteration 0 seems missing.
Q3: According to Appendix B and Figure 5, the authors use the likelihood of the proof under the policy as a measure of difficulty. To what extent is the distinction of the difficulty between the log-likelihood of -13 and -1? Could you please provide some cases for each?
Q4: Figure 2 shows diminishing gains in conjecture and policy iteration 4. Does it mean approaching the saturated performance? How to decide the number of iterations?
Q5: How many evaluated theorems are in the Natural Number Game dataset?

**Limitations:**

The authors have adequately addressed the limitations.

---

> ### Author Rebuttal · Authors · 2024-08-07
>
> We thank the reviewer for the encouraging review of our work! We provide clarifications below, and would be happy to answer any questions that remain.
>
> > Further justification of using log-likelihood as the main evaluation in this paper.
>
> - It can be seen as a sort of continuous version of the success rate. We found that the more likely the proof is under a policy, the faster MCTS reaches it. Thus, whether MCTS finds it within the budget depends on the search cut-off, whereas the likelihood doesn't (even two policies that both fail, or both succeed, on the same problem can still be differentiated by the likelihood).
> - Although closely related, it is much faster to compute. While proof search can take 1-4 minutes depending on the problem and with the search budget we used, the likelihood can be computed in a second. This makes larger scale analyses feasible, such as the one in Figure 3 (> 50k proof-policy evaluations).
>
> > How does this method facilitates solving dataset such as miniF2F and mathlib?
>
> We emphasize that the focus of our contribution was mathematical discovery and self-improvement, starting only from axioms, but crucially no pre-training data. This is fundamentally different from approaches that rely on massive pre-training on existing problems, which is common to the methods typically evaluated on mathlib or minif2f. While we hope that agents trained in a setup like ours will eventually be able to rediscover much of mathlib and solve problems from minif2f, there are still scalability challenges that require further research (such as in accumulating a library, discussed above). If those are overcame, then this would be one path to impacting those benchmarks. Another path would be to investigate a setup similar to what we propose, but starting with a pre-trained LLM. Conjecturing can help expand the existing training data, and potentially improve performance on the benchmarks. However, it's still unclear whether this would generalize to domains beyond those seen in pre-training, such as those already on mathlib. Our goal was to not have this dependency.
>
> > Q1: How to guarantee f_C soundness and completeness?
>
> Completeness is given by the fact that the completion engine expansion rules (L193-201) consider all of the typing rules in Peano (which are few). Thus, given any valid conjecture c, the choices that have to be made to generate c have to fall in one of these cases. Soundness relies on the soundness of unification in Peano, but is basically the fact that at each step we either directly give out an object from the context from the desired type (which is sound), or give the result of a function that Peano believes can produce the desired type (by unifying the type with the result type of the function).
>
> > Q2: In Figure 2, propositional logic, the search baseline policy at iteration 0 seems missing.
>
> It is there, but almost coincides with iteration 1. We generally found little improvement in the policy in propositional logic after a single iteration, but it starts to pick up later.
>
> > Q3: According to Appendix B and Figure 5, the authors use the likelihood of the proof under the policy as a measure of difficulty. To what extent is the distinction of the difficulty between the log-likelihood of -13 and -1? Could you please provide some cases for each?
>
> Please refer to the global response for additional examples in each domain with their likelihoods. We will include these in the appendix.
>
> > Q4: Figure 2 shows diminishing gains in conjecture and policy iteration 4. Does it mean approaching the saturated performance? How to decide the number of iterations?
>
> Yes, the policies start to converge at that point. We believe that this is due to the main limitation we highlighted in Section 5: our agent still can't come up with new definitions, or reuse past theorems. As a result, the way the agent finds to produce harder conjectures is to make the statements more convoluted. But in human mathematics, even deep, interesting theorems tend to have short statements (posed in terms of the right high-level definitions). Our main contribution is to set up a self-improvement pipeline where future work can tackle the open problem of understanding how to build towards "interesting" deeper theorems. If this is achieved, then in principle there would be no limit to how many iterations one could run for, as the agent would be able to keep raising the level of abstraction of the theorems it discovers (which our current agent does not yet do).
>
> > Q5: How many evaluated theorems are in the Natural Number Game dataset?
>
> The whole game (in the website) contains 83 unique theorems. Since in its current form our method doesn't yet accumulate a library with its previously proved theorems, it can only plausibly prove the levels that only refer to the axioms (rather than previous lemmas), which leaves 10 theorems spread across the first 3 "worlds" in the game (Tutorial, Addition, Multiplication). We included all these problems in Appendix E.
>
> Thank you again for the comments and questions! We'd be happy to answer any further questions, or discuss these in more depth.

---

> > ### Comment · Reviewer_6DqB · 2024-08-13
> >
> > Thanks for the rebuttal. I keep the score.

---

### Official Review · Reviewer_viHR · 2024-07-13

**Soundness:** 3
**Presentation:** 3
**Contribution:** 3
**Rating:** 7
**Confidence:** 4

**Summary:**

The authors investigate a novel setting for ML-based formal theorem proving, where the proving agent in addition to learning to prove theorems at the same time learns to propose conjectures. The process is composed into a self-improving feedback loop that starts just from axioms and continues to prove increasingly difficult self-generated conjectures. The hindsight relabeling method is introduced that extracts training data points even from the failed proof searches, which improves the learning process.

**Strengths:**

Formal theorem proving is a great challenge for testing and developing AI approaches. In contrast to natural language mathematical settings, the formal environment provides grounding not allowing for any mistakes by the ML agent.

The topic of conjecturing is an interesting one and at the same time very much under-explored. Moreover, combining proving and conjecturing is novel and, as far as I know, was not researched before.

The proposed setting is arguably quite simple, but as this is one of the first studies of this kind it is justifiable and actually beneficial -- it allows to more precisely control and understand the whole learning process, which may inform some follow-up studies.

The authors provide code for the Peano environment and for reproducing the learning experiments.

**Weaknesses:**

As the presented setting is limited (simple conjectures generated, no mechanism for abstracting reusable lemmas), it is not clear to what extent the proposed methodology will transfer to fully-fledged formal proving environments like Lean, Coq, or other ITPs. (The authors are aware of this limitation and they mention it in Section 5.)

The presented experiments are small: only five iterations of conjecturing-proving loop are run.

Some hyper-parameters of the experimental setup are fixed in an ad hoc manner without performing any grid searches (n. of conjectures generated per iteration, n. of expansions in the MCTS). It would perhaps be good to measure the effect of some of these parameters.

It would be interesting to see more metrics tracked across the learning iterations for better insights about the process, for instance:
- the average length/syntactic complexity of the generated conjectures,
- the average length of a proof per iteration,
- the duplication rate between consecutive batches of generated conjectures.

Some details of the method are not specified clearly (see my questions below).

Minor:
- evaluate as -- evaluated as (the caption of Fig. 2)

**Questions:**

How the conjecture generation is conditioned on the difficulty level?

How do you tokenize formulas?

How exactly do you calculate the average proof log-likelihood (Figure 2):
- how do you incorporate the failed proof attempts into the average?
- what is the set of conjectures you compute the average for: is it across a new batch of 200 conjectures, or the old ones (which were used for training the prover) are also used?
- are the conjectures from hindsight relabeling taken into the average here?

Is it correct that the lines in Fig. 2, right, for the policies from iterations 0 and 1 are completely overlapping?

Why do you display the proof log-likelihood instead of perhaps more interpretable metrics like the proving success rate for a batch of new conjectures or the average length of a proof search / a proof size?

Why group theory is missing from the evaluation presented in Fig. 4?

Is the success rate in Fig. 4 computed independently per each iteration, or rather cumulatively (taking union of theorems proved in all the past iterations)?

Could you provide a list of requirements to create a Python environment for running the supplemented code? I would like to test it, but there is no detailed installation instructions.

**Limitations:**

The authors correctly identify the major limitations of their approach.

---

> ### Author Rebuttal · Authors · 2024-08-07
>
> We thank the reviewer for the through and encouraging assessment of our work! We provide clarifications below, and would be happy to answer any questions that remain.
>
> > How the conjecture generation is conditioned on the difficulty level?
>
> Once we attempt to prove a conjecture, we obtain a difficulty evaluation (failed, easy or hard, where easy and hard are relative to the batch of conjectures that were attempted). Using this feedback, we then generate training examples (strings) where the difficulty comes first, followed by the statement of the conjecture. For example (examples generated from the first conjecturing iteration of a run on propositional logic, after running proof search on this batch):
>
> ```
> "Conj:(fail) false"
> "Conj:(triv) (iff (not (not (not false))) false)"
> "Conj:(hard) [('a0 : (or false false)) -> false]"
> ```
>
> All these conjectures were first sampled by conditioning on "Conj:(hard) " (the conjecturer starts untrained, so initially that doesn't mean anything to it). Then, after training on the strings above, in addition to the examples from proof search, we sample again the next batch of conjectures.
>
> > How do you tokenize formulas?
>
> We used character-level tokenization. Since we assumed no initial data to train a BPE tokenizer, we found this to be the simplest, more general choice.
>
> > How exactly do you calculate the average proof log-likelihood (Figure 2):
>
> A proof p corresponds to a sequence a_1, …, a_n of actions in the search tree. For the proof log-likelihood, we average the log-likelihood of each action being taken at the corresponding state. So if the starting state is s_0, that is the average of log p(a_1 | s_0), log p(a_2 | s_1), and so on. Each s_k is obtained by taking the action in the proof from the previous state. Each of these probabilities is a readout from the policy LM. The closer this probability is to 1, the more likely is this proof for the LM; correspondingly, the faster MCTS will explore these paths and find this proof during proof search.
>
> > how do you incorporate the failed proof attempts into the average?
>
> In Figure 2 we only analyze conjectures that were proved (4.1, L302). We cannot compute the likelihood above if we indeed find a proof (since p appears in the formula). However, given one proof, we can evaluate its likelihood under any policy, even if previous policies assign much lower likelihood to it (and thus would take longer search, or time out, in finding it). We show and discuss the fraction of proved conjectures at each iteration in Appendix C.
>
> > what is the set of conjectures you compute the average for: is it across a new batch of 200 conjectures, or the old ones (which were used for training the prover) are also used?
>
> Only new conjectures that were proposed at each iteration. They are guaranteed to be (at least syntactically) different from previous ones, since we reject samples that have been seen in the past.
>
> > are the conjectures from hindsight relabeling taken into the average here?
>
> No, only the conjectures in the batch are.
>
> > Is it correct that the lines in Fig. 2, right, for the policies from iterations 0 and 1 are completely overlapping?
>
> That's right, they were nearly indistinguishable.
>
> > Why do you display the proof log-likelihood instead of perhaps more interpretable metrics like the proving success rate for a batch of new conjectures or the average length of a proof search / a proof size?
>
> There are two main reasons for using the likelihood compared to success rate:
> - It can be seen as a sort of continuous version of the success rate. We found that the more likely the proof is under a policy, the faster MCTS reaches it. Thus, whether MCTS finds it within the budget depends on the search cut-off, whereas the likelihood doesn't (even two policies that both fail, or both succeed, on the same problem can still be differentiated by the likelihood).
> - Although closely related, it is much faster to compute. While proof search can take 1-4 minutes depending on the problem and with the search budget we used, the likelihood can be computed in a second. This makes larger scale analyses feasible, such as the one in Figure 3 (> 50k proof-policy evaluations).
>
> > Compared to proof size, the main advantage is that the likelihood takes into account the branching factor. There are long proofs that are nonetheless easy because there aren't many options at each step (e.g., most are 'intro').
> Is the success rate in Fig. 4 computed independently per each iteration, or rather cumulatively (taking union of theorems proved in all the past iterations)?
>
> Independently.
>
> > Could you provide a list of requirements to create a Python environment for running the supplemented code? I would like to test it, but there is no detailed installation instructions.
>
> We apologize, our release script ignored .txt files, but that included requirements.txt. Here is the content of this file (in `learning`):
>
> ```
> altair
> bottle
> coloraide
> hydra-core
> maturin
> numpy
> omegaconf
> redis
> rq
> sympy
> torch
> tqdm
> transformers
> wandb
> celery
> ```
>
> maturin installs an executable (called maturin) that allows you to compile and install the Peano Python package easily. You can go to environment/ and run maturin develop –release to both build and automatically "pip install" the package in your local environment (after that, `import peano` should work from Python). We will expand broadly on our tutorial in the public code release.
>
> For an additional analysis of the generated conjectures across iterations, please refer to our global response.
>
> We will update the paper to incorporate these descriptions that were missing or confusing. We again thank the reviewer, and would be happy to discuss or clarify further!

---

> > ### Comment · Reviewer_viHR · 2024-08-14
> >
> > Thank you for your rebuttal, it addressed my questions very well! I keep my positive score.

---

### Official Review · Reviewer_8wJE · 2024-07-16

**Soundness:** 3
**Presentation:** 3
**Contribution:** 2
**Rating:** 5
**Confidence:** 3

**Summary:**

This paper presents a new method for training LLM agents in mathematical reasoning, starting only from basic axioms. The main idea is to make the agent learn two skills together: coming up with hard math problems (conjecturing) and solving them (theorem proving). The authors use a language model to do both tasks, and they introduce clever ways to generate valid math statements and learn from failed attempts at proofs. They test their method on three areas of math: logic, arithmetic, and group theory. The results show that the agents get better at both making harder problems and solving them over time. Importantly, the agents can also solve some human-written math problems they weren't directly trained on.

**Strengths:**

1. The paper introduces a new way to train LLMs for math, starting only from basic rules (axioms). It makes the LLM learn to create hard math problems and solve them at the same time, which is different from other methods that use lots of human-made math examples.
2. The authors use smart techniques to make their method work well, like special ways to generate valid math statements and learn from failed attempts. They test their method on three different areas of math, showing it works in various situations. The paper explains these ideas clearly, though some parts might be hard for people who don't know much about certain math topics.
3. This work could lead to AI systems that can do math research on their own, without needing human-made examples. This might help discover new math ideas in areas people haven't explored much. It's also important for making LLM that can think and create on its own, not just follow human instructions.

**Weaknesses:**

1. The current approach does not accumulate a growing library of proven theorems, which limits its ability to tackle more complex mathematical problems efficiently. This restriction to essentially "cut-free" proofs could become a significant bottleneck as the complexity of the target domain increases.
2. The paper doesn't show how this new way compares to other neural ATP methods. This makes it hard to know how good it really is.
3. The authors didn't try their model on standard math problem sets like mathlib or mini-f2f, which many other neural ATP methods use. This makes it hard to compare their results to other research.

**Questions:**

1. Did the authors consider evaluating the system on widely-used benchmarks like mathlib or mini-f2f? If not, what are the main challenges in adapting the approach to these benchmarks?
2. How well does an agent trained in one mathematical domain (e.g., propositional logic) generalize to another (e.g., arithmetic)?

**Limitations:**

Yes

---

> ### Author Rebuttal · Authors · 2024-08-07
>
> We thank the reviewer for the through review and encouraging comments about our work, especially about not relying on human-made examples. We answer the questions below, but are happy to engage further.
>
> > The current approach does not accumulate a growing library of proven theorems, which limits its ability to tackle more complex mathematical problems efficiently. This restriction to essentially "cut-free" proofs could become a significant bottleneck as the complexity of the target domain increases.
>
> That's correct, we acknowledge this current limitation in the paper. We note that accumulating a library by itself is not technically challenging – the main open research problem here is to determine what theorems are worth adding to the library. This might require some metric for the interestingness, or usefulness, of a theorem. These are important but still understudied topics in mathematical discovery, as most current work on AI for mathematics only focuses on solving a target set of problems, not discovering new theorems autonomously. Our work sets up an experimental foundation for future work to explore these metrics and empirically observe what libraries they end up discovering.
>
> > The paper doesn't show how this new way compares to other neural ATP methods. This makes it hard to know how good it really is.
>
> Our work contributes both a new problem setting where an agent bootstraps itself from only the axioms of a domain, and a pipeline for learning in this setting. One of the components of this pipeline is the prover (ATP) agent, but the pipeline is agnostic to the specific neural ATP method that the prover employs. Our specific ATP is most similar to Holophrasm [1], using MCTS with learned value and policies, as has become standard since then. But any other proof search method from prior work, like HTPS, could also work with our pipeline, complementing the other components (conjecturing, and the outer learning loop). Thus, our pipeline is not a direct competitor to other neural ATP methods, but rather a novel method for bootstrapping one using no pre-existing training data. To the best of our knowledge, our system is the first to learn entirely from self-generated conjectures, without requiring human training data. This is our main contribution.
>
> > Did the authors consider evaluating the system on widely-used benchmarks like mathlib or mini-f2f? If not, what are the main challenges in adapting the approach to these benchmarks?
>
> We emphasize that the focus of our contribution was mathematical discovery and self-improvement, jointly learning to conjecture and prove theorems starting only from axioms, but crucially no pre-training data. This is fundamentally different from approaches that rely on massive pre-training on existing problems, which is common to the methods typically evaluated on mathlib or minif2f.
> While we hope that agents trained in a setup like ours will eventually be able to rediscover much of mathlib and solve problems from minif2f, there are still scalability challenges that require further research (such as in accumulating a library, discussed above) before benchmarks like minif2f become within reach. However, if we can overcome those challenges, our approach in principle can work in new mathematical domains, whereas it's still not clear how methods that require massive pre-training to do well will be capable of that generalization.
>
> > How well does an agent trained in one mathematical domain (e.g., propositional logic) generalize to another (e.g., arithmetic)?
>
> We attempted to initialize an agent for one domain using the last checkpoint of another to evaluate transfer, but did not observe any improvements. We will note this in the appendix. We believe that this is just due to overfitting, since the three domains we tested are very different from each other (for instance, axiom names, types, etc, are all different, thus completely unseen when the agent starts in the new domain). We believe that bootstrapping on a significantly wider spanning set of domains, coupled with library learning, might be enough to observe transfer.
>
> We also note that we provided more examples of conjectures and proofs in the global response.
>
> Thanks again for the review! We'd be happy to answer further questions.

---

> > ### Comment · Reviewer_8wJE · 2024-08-13
> >
> > Thank you for the rebuttal, which solves some of my concerns. I'm happy to raise my score.

---

### Author Rebuttal · Authors · 2024-08-07

We thank all reviewers for the detailed evaluations of our submission. We appreciate the general comments on the novelty of our conjecturing and proving loop. One common request was to show more examples of conjectures and proofs as they evolve during training. We provide examples here (all *sampled* conjectures, not from hindsight). Due to space, we show iterations 0, 2 and 4 for groups, with some complete proofs, and summaries for other domains.

# Groups

## Iteration 0

**Avg. proof length**: 2.67 steps

**Avg. proof length on 'hard' conjectures**: 3.67 steps

### Hardest 3 problems:

Conjecture: `[('a0 : (= id (op id (op id (op (op (op id id) (inv id)) (inv (op id id))))))) -> (= (op id (inv id)) id)]`  (Proof Logprob: -8.51928925095947, 4 steps)

Proof:
```
  show (= (op (inv id) id) id) by inv_l.
  show (= (op (inv id) id) (op id (inv id))) by op_comm.
  show (= (op id (inv id)) id) by rewrite.
  intro _ : [('a0 : (= id (op id (op id (op (op (op id id) (inv id)) (inv (op id id))))))).
```

Conjecture: `(= id (op (inv id) id))`  (Proof Logprob: -7.6116456751792665, 3 steps)

Proof:
```
 show (= (op id id) id) by id_l.
 show (= (op (inv id) id) id) by inv_l.
 show (= id (op (inv id) id)) by eq_symm.
```
Conjecture: `[('a0 : G) -> ('a1 : G) -> (= 'a0 (op id 'a0))]`  (Proof Logprob: -7.2510264460057865, 4 steps)

### Easiest 3 problems:

Conjecture: `[('a0 : (= id id)) -> ('a1 : (= (op (op id id) id) id)) -> (= id id)]`  (Proof Logprob: -2.347079877108669, 2 steps)

Conjecture: `[('a0 : (= id id)) -> (= id id)]`  (Proof Logprob: -1.5468491897693764, 1 steps)

Conjecture: `(= id id)`  (Proof Logprob: -1.2030829999623922, 1 steps)

## Iteration 2

**Avg. proof length**: 4.10 steps

**Avg. proof length on 'hard' conjectures**: 6.88 steps

### Hardest 3 problems:

`[('a0 : G) -> ('a1 : (= id (inv (op id id)))) -> (= (inv (op id id)) (inv (op (inv (op id (inv (op id id)))) id)))]`  (Proof Logprob: -6.0811389550124035, 5 steps)

`(= (op (op id id) (op id id)) (op id (op id id)))`  (Proof Logprob: -5.707431809481172, 3 steps)

`[('a0 : (= (inv id) id)) -> ('a1 : G) -> ('a2 : G) -> ('a3 : G) -> ('a4 : G) -> ('a5 : (= 'a4 (inv id))) -> (= 'a4 (inv (inv (inv id))))]`  (Proof Logprob: -5.4084035606314, 9 steps)

### Easiest 3 problems:
`[('a0 : (= id (op id (op (inv id) (op id id))))) -> ('a1 : G) -> ('a2 : (= id 'a1)) -> (= id 'a1)]`  (Proof Logprob: -0.2945812885670408, 3 steps)

`[('a0 : (= (op (inv id) (op id id)) (inv id))) -> ('a1 : G) -> ('a2 : (= 'a1 'a1)) -> (= 'a1 'a1)]`  (Proof Logprob: -0.20977646226311422, 3 steps)

`[('a0 : G) -> ('a1 : G) -> ('a2 : G) -> ('a3 : G) -> ('a4 : G) -> ('a5 : (= 'a4 'a4)) -> ('a6 : G) -> ('a7 : (= 'a2 (op 'a0 (op (inv 'a4) 'a4)))) -> (= 'a4 'a4)]`  (Proof Logprob
: -0.17343407015213588, 8 steps)


## Iteration 4

**Avg. proof length**: 5.00 steps

**Avg. proof length on 'hard' conjectures**: 6.10 steps

### Hardest 3 problems:

`[('a0 : G) -> ('a1 : (= id (op (inv 'a0) (inv 'a0)))) -> (= (op id id) (op id (op id (op id (op (inv 'a0) (inv 'a0))))))]`  (Proof Logprob: -9.726059012187891, 9 steps)

Proof:
```
 intro x : G.
 intro x0 : (= id (op (inv x) (inv x))).
 show (= (op id (op id (op (inv x) (inv x)))) (op id (op (inv x) (inv x)))) by id_l.
 show (= (op id (op (inv x) (inv x))) (op id (op (inv x) (inv x)))) by rewrite.
 show (= (op (inv x) (inv x)) (op (inv x) (inv x))) by rewrite.
 show (= (op id (op (inv x) (inv x))) (op id (op id (op (inv x) (inv x))))) by eq_symm.
 show (= (op (inv x) (inv x)) id) by eq_symm.
 show (= (op id id) (op id (op id (op (inv x) (inv x))))) by rewrite.
 show (= (op id id) (op id (op id (op id (op (inv x) (inv x)))))) by rewrite.
```

`[('a0 : G) -> (= (op id (op (inv 'a0) (op 'a0 'a0))) (op id (op (op 'a0 'a0) (inv 'a0))))]`  (Proof Logprob: -7.7850161602701835, 6 steps)

`[('a0 : (= (inv id) (inv (op id (inv id))))) -> (= (inv id) (op id (inv (op id (inv (op id (inv id)))))))]`  (Proof Logprob: -7.032775252968119, 5 steps)

### Easiest 3 problems:

`[('a0 : G) -> ('a1 : G) -> (= (op id (op id (op 'a1 (inv id)))) (op id (op 'a1 (inv id))))]`  (Proof Logprob: -1.017160122076225, 3 steps)

`[('a0 : G) -> (= (op id (op 'a0 (op id (op id 'a0)))) (op 'a0 (op id (op id 'a0))))]`  (Proof Logprob: -0.8543304965071575, 2 steps)

`[('a0 : G) -> ('a1 : (= 'a0 (op (op 'a0 (inv 'a0)) 'a0))) -> (= 'a0 (op (op 'a0 (inv 'a0)) 'a0))]`  (Proof Logprob: -0.08850505636260465, 2 steps)

# Prop. Logic

**Avg. proof length**: 2.75 -> 4.14 -> 4.21 steps

**Average proof length on 'hard' conjectures**: 5.75 -> 7.67 -> 7.78 steps

**Top hard conjecture in iteration 0 vs 4**: `[('a0 : false) -> false]` (1 step) vs ` (or (not (and (not false) (not (or (not false) (or false false))))) (or (not (not (or (or false false) (not (or false false))))) false))` (11 steps)

# Arithmetic

**Avg. proof length**: 2.36 -> 2.50 -> 3.35 steps

**Average proof length on 'hard' conjectures**: 4.00 -> 4.62 -> 5.50 steps

**Top hard conjecture in iteration 0 vs 4**:
`(= (* z z) (* (* (* z z) (* (s (* z z)) (s z))) z))` (4 steps) vs `[('a0 : (= (s (+ z (+ z z))) z)) -> (= z (s (+ z (+ z (s (+ z z))))))]`  (7 steps)

---

### Decision · Program_Chairs · 2024-09-25

**Decision:**

Accept (oral)

**Comment:**

The paper provides a novel scheme for generating mathematical conjectures and proving them when the input is the theory axioms only.
All reviewers acknowledged that this is a challenging and highly interesting task and that the approach is quite pioneering.  The critic by reviewers is centered on the small scale of experiments, and issues of non-scalability. The authors provided detailed rebuttals that was well received by the reviewers (and one raised its score). Overall,  all reviewers were quite positive.